# Contributions of distemper control and habitat expansion to the Amur leopard viability

Dawei Wang [1], Francesco Accatino[2], James L. D. Smith[3] & Tianming Wang [1✉]

The Amur leopard (*Panthera pardus orientalis*) is a critically endangered top predator that struggles on the brink of extinction due to threats such as canine distemper virus (CDV), habitat loss, and inbreeding depression. Here we develop a viability analysis metamodel that combines a traditional individual-based demographic model with an epidemiological model to assess the benefits of alternative population management actions in response to multiple distinct threats. Our results showed an extinction risk of 10.3%-99.9% if no management actions were taken over 100 years under different levels of inbreeding depression. Reducing the risk of CDV infection in Amur leopards through the low-coverage vaccination of leopards and the management of sympatric domestic dogs could effectively improve the survival probability of the leopard population, and with habitat expansion added to these management measures, the population expanded further. Our findings highlight that protecting the Amur leopard necessitates a multifaceted synergistic effort, and controlling multiple threats together may significantly escalate overall viability of a species, especially for small-isolated threatened population. More broadly, our modeling framework could offer critical perspectives and scientific support for conservation planning, as well as specific adaptive management actions for endangered species around the world.

---

[1] Ministry of Education Key Laboratory for Biodiversity Science and Engineering, NFGA Key Laboratory for Conservation Ecology of Northeast Tiger and Leopard & College of Life Sciences, Beijing Normal University, 100875 Beijing, China. [2] UMR SADAPT, INRAE, AgroParisTech, Université Paris-Saclay, 22 place de l'agronomie, CS 80022, 91120 PALAISEAU Cedex, France. [3] Department of Fisheries, Wildlife and Conservation Biology, University of Minnesota, St. Paul, MN 55108, USA. ✉email: wangtianming@bnu.edu.cn

large carnivores, the world's most revered and iconic animals, struggle on the brink of extinction due to a variety of threats, such as habitat loss and fragmentation, scarcity or lack of prey, diseases, and human interference[1–3]. Living in low-quality fragmented habitat patches is challenging for these species because of their large habitat and high energy requirements[4,5] and diseases make these species more vulnerable[6,7]. As a result, the population size and distribution range of many large carnivores such as tigers (*Panthera tigris*) and lions (*P. leo*) have recently shrunk to a historically low level[2,8] When the population size drops below a certain threshold, the species might likely go extinct quickly (which is the Allee effect)[9,10], and the loss of the top carnivores might also initiate unexpected cascade effects throughout the entire food chain[11].

Amur leopard (*P. pardus orientalis*) is, like many large carnivores, in a seriously compromised situation; it is one of the rarest subspecies in the feline family and listed as "critically endangered" in the IUCN Red List of Threatened Species[12]. At present, the Amur leopards are confined to a forest habitat of approximately 9,000 km² in the Land of Leopards National Park in the southwestern Primorye Province of Russia and the neighboring Northeast Tiger and Leopard National Park in the Jilin and Heilongjiang provinces of China[13,14]. Its habitat consists of the northern temperate coniferous and broadleaved mixed forests, where it is nearly the largest feline, with a body size inferior only to the Amur tiger (*P. t. altaica*), which is also at risk of extinction[5].

Like many isolated populations, this population is also threatened by environmental stochasticity, inbreeding depression, infectious diseases, poaching, and other factors[15]. The canine distemper virus (CDV) is considered a new threat to the Amur leopard population. CDV is a multihost single-stranded RNA virus, regarded as the pathogen of greatest threat to large felids worldwide[16]. It has a nearly-global distribution[17], with outbreaks confirmed to be related to the declines or near extinction of several wild animal populations, including Serengeti lions[18], Sumatran tigers (*P. t. sumatrae*)[19], Ethiopian wolves (*Canis simensis*)[20], and Santa Catalina Island fox (*Urocyon littoralis catalinae*)[21]. CDV has been spreading within the Amur leopard population since 1993, and in 2015 a wild Amur leopard in Russia was diagnosed with a CDV infection[22]. The leopard can prey on various virus hosts (free-ranging domestic dogs and small sized carnivores such as the Asian badger *Meles meles*, red fox *Vulpes vulpes*, and leopard cat *Prionailurus bengalensis*) that act as infection pools[15].

At present, the main strategy for the prevention and control of CDV in endangered species is the control of the infection pool (as concluded for tigers by Gilbert et al. (2015))[23]. For the Amur leopard, because the management of wild animals poses challenges, the focus was on the control of domestic dogs. However, in addition to dogs, small and medium-sized wild carnivores still play an important role as a source of infection for endangered species[6,24]. Therefore, simply controlling or vaccinating free-ranging domestic dogs is not enough to protect Amur leopards. Another possible control measure is to directly vaccinate endangered species populations, which has been controversial in the past due to safety and operational difficulties[25]. Low-coverage vaccination (i.e., vaccinating a small percentage of the population) has been proven to be effective in preventing the extinction of Ethiopian wolves due to the rabies virus[26] and, via model simulation, was shown to be a possibility to reduce the extinction probability of the Amur tiger population due to CDV[6].

Additionally, due to the urban development between southwest Primorye and Sikhote-Alin in Russia, the spread of the Amur leopard population in Russia has been limited[27]; in contrast, there is a vast potential habitat on the Chinese side[5,28]. Currently, the Northeast Tiger Leopard National Park in China and the Land of Leopard National Park in Russia have provided area conditions for the maintenance of the tiger and leopard populations (with a total area of nearly 18,000 km²). This measure provides a great opportunity for developing the population of the Amur leopard; however, there is no research available to assess the benefits of this measure for the Amur leopard population.

Population viability analysis (PVA) is a modeling technique for analyzing the dynamics of endangered species[29,30], and helps identifying determinants of population decline for effective management recommendations[31]. Vortex software package has a wide range of applications in PVA, which has been used to evaluate the survival status and different management strategies for the conservation of endangered species, such as the griffon vulture (*Gyps fulvus*), fennec fox (*V. zerda*), and mountain lion (*Puma concolor*)[32–34]. In addition, Vortex incorporates inbreeding depression (the organism's reduced ability to survive as a result of the inbreeding of related individuals), which is very important for small-sized populations like the object of our study. However, PVA has never been applied to the Amur leopard.

The purpose of this study was to explore three management measures to lessen the impact of CDV on the Amur leopard population: domestic dog control, Amur leopard low-coverage vaccination, and habitat expansion. We constructed a PVA metamodel including not only the population dynamics model of Amur leopard, but also the CDV epidemiology. First, to explore the effects of CDV on the viability of the Amur leopard population, we constructed a separate CDV epidemiological model by using the software Outbreak, a powerful tool for simulating the disease dynamics in populations[35]. Second, we used Vortex to build the population dynamic model. Last, we used a metamodel approach to combine the epidemiological and population dynamics model using Metamodel Manager[36,37]. We simulated and analyzed the population viability of the Amur leopard over the next 100 years and the influence of the three management measures, assuming different combinations of management measures and different inbreeding depression scenarios. In addition, a sensitivity analysis was performed to identify the most important factors affecting population survival. As far as we know, this is the first study of PVA for the Amur leopard that incorporates an independent epidemiological model. The results of our analysis can inform the management policy for the Amur leopard's persistence.

## Results

### The simulations of baseline and alternative management actions.
We represented the trajectories of the population size for all the simulated management actions. To improve the visualization, we represented the incertitude bars only for the baseline and the combination of the three management alternatives combined together (A + B + C) (Fig. 1, column on the left). Other metrics calculated from the simulations are given in Fig. 2. In the A + B + C scenario, the best population growth was obtained. The standard deviation increased with time in both the baseline scenario and in the A + B + C scenario; the greater number of surviving leopards in the A + B + C scenario increased the randomness of the model and the standard deviation range of A + B + C was higher and wider than in the baseline scenario (Fig. 1).

All trajectories showed a small increase in years 9-13, related to the initial setting of the age structure and the sex ratio, having little effect on the longer-term population trend. Comparing baseline with the trajectory of the other scenarios, it was evident that the Amur leopard population faced a greater risk of extinction when no conservation measures were implemented

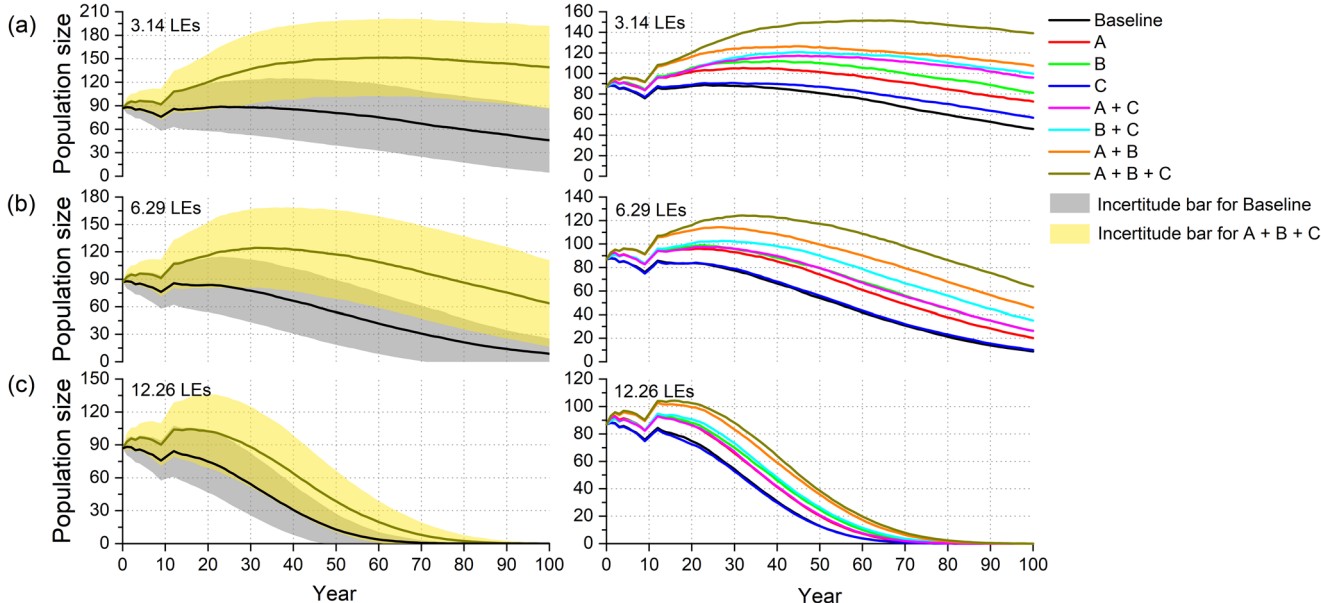

**Fig. 1 Simulated time trajectories of population size over 100 years for the Amur leopard population on the China-Russia border under different management alternatives and combinations, based on three values of inbreeding depression.** Scenario A: controlling the domestic dogs around the habitat; Scenario B: low-coverage vaccination of leopards (6 leopards each year); Scenario C: expanding habitat for leopards. The data for the baseline and (A + B + C) scenarios are expressed as the mean ± standard deviation in the left column (see the incertitude bars), and all the average trajectories are represented in the right column. Average trajectories are calculated out of 1000 model runs. **a–c** Population trajectory under the inbreeding depression scenarios with lethal equivalents of 3.14, 6.29, and 12.26, respectively.

in all three cases of inbreeding depression (Fig. 1). Even under the mildest inbreeding depression scenario (3.14 LEs), the extinction probability in the baseline scenario reached 20.2% (Fig. 2b, Table S6), and the mean stochastic population growth rate was −0.011, while in the scenario with 12.26 LEs, the extinction probability reached 99.9% (Fig. 2f), with a rate of population decline more than five times than in the 3.14 LEs scenario (Table S6).

Almost all management interventions (or combinations thereof) showed a reduction of the leopard population at the end of the time horizon, but some performed better than others, with some showing a population increase at the end of the simulation (Fig. 1a). We found that, under the inbreeding depression of 3.14 LEs and 6.29 LEs, the low-coverage vaccination was the most effective across the three values of inbreeding depression, followed by the control of free-ranging domestic dogs (Fig. 1). With these two management alternatives, population size, genetic diversity and survival probability at year 100 were better than in the baseline scenario (Fig. 2, Table S6), but the population decline trend over 100 years did not change (Fig. 1). Compared to the baseline scenario, the 100-year population development trend in the scenario of habitat expansion did not improve significantly. However, at an inbreeding suppression of 12.26 LEs, all measures implemented did not prevent the population from becoming extinct within 100 years (Fig. 1c, Table S6).

Among the pairs of management alternatives, all combinations of measures resulted in a larger population size than the initial population size after 100 years under the mildest inbreeding depression of 3.14 LEs, but at 6.29 LEs and 12.26 LEs, the combination of measures did not change the trend of population decline. Among them, the combination of dog control and habitat expansion (A + C) exhibited the worst performance, and the trajectory of population size was similar to the trajectory for the low-coverage vaccination implemented alone under the inbreeding depression of 6.29 LEs (Fig. 1). At the end of the simulation,

this combination had the lowest population size, genetic diversity and probability of survival among all combinations of management alternatives (Fig. 2, Table S7). In contrast, population size trends were better in the management alternative combinations that included the low-coverage vaccination.

The best combination was A + B + C: for the 3.14 LEs, the population achieved maximum positive growth, with a mean population growth rate of 0.008, genetic diversity of 0.86, survival probability of the population reaching 99% and population size increasing to 139 within 100 years (Fig. 1a, Fig. 2a, b, Table S7); for 6.29 LEs, the population largely maintained its size at 64, with a mean population growth rate of −0.006, genetic diversity of 0.82, and population survival probability of 90% (Fig. 1b, Fig. 2c, d, Table S7); for 12.26 LEs, the mean population decline rate was −0.046, and the population was extinct after 100 years (Fig. 1c, Fig. 2e, f, Table S7).

In conclusion, the population trajectory eventually declined in the majority of scenarios, except the combinations of management alternatives in the case of 3.14 LEs inbreeding depression, which showed population growth. The best performing of these was the combination of three measures. When the level of inbreeding depression was severe, all management strategies failed to alter the fate of the population from extinction.

**Low-coverage vaccination**. Low-coverage vaccination of leopards positively impacted population size (Fig. 3). Increasing the number of vaccinated individuals per year, the population survival probability increased for all inbreeding depression values. However, this increase was not linear; in the case of 3.14 LEs, when the number of individuals vaccinated increased from 0 to 2, the population survival probability increased by 8.8% and when the number of individuals vaccinated increased from 8 to 10, it increased by 0.2% (Fig. 3b). In the case of 6.29 LEs, the corresponding increases were 48.0% and 10.6%, respectively (Fig. 3d).

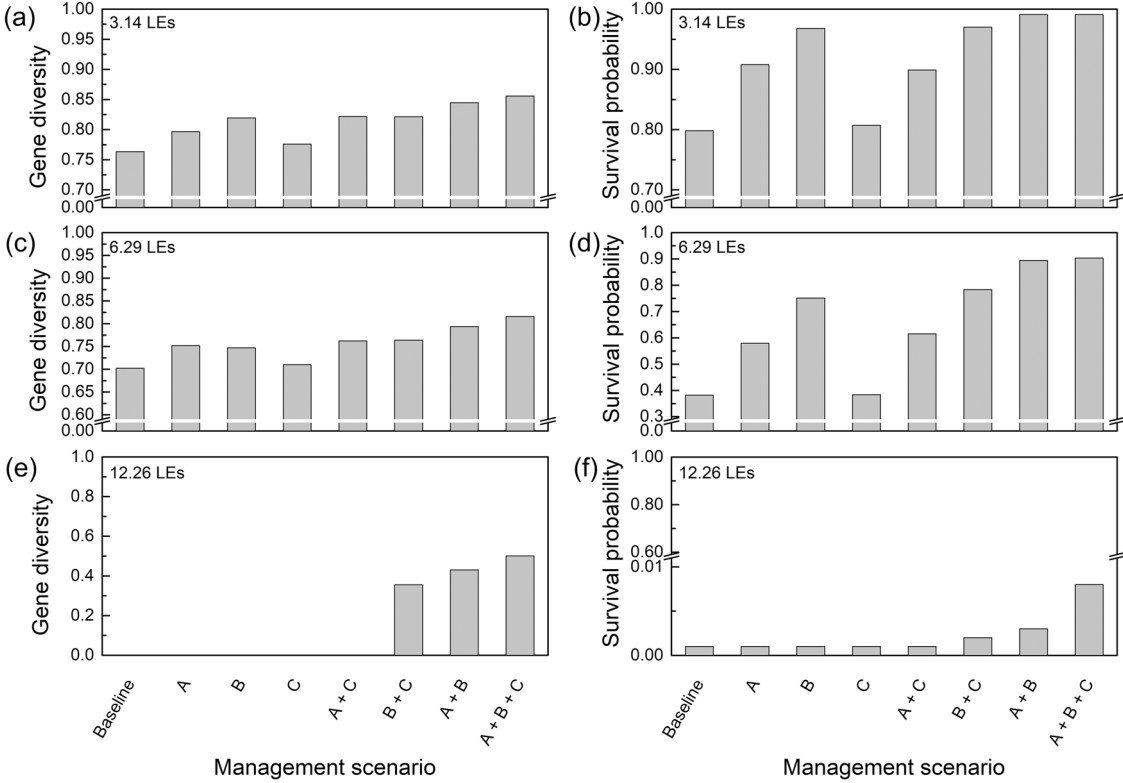

**Fig. 2 Estimates of gene diversity and survival probability at the 100th year for the Amur leopard population under different management alternatives and combinations for the three values of inbreeding depression.** Scenario A: controlling the domestic dogs around the habitat; Scenario B: low-coverage vaccination of leopards (6 leopards each year); Scenario C: expanding habitat for leopards. **a, c, e** Gene diversity of Amur leopard population at the 100th year under the inbreeding depression scenarios with lethal equivalents of 3.14, 6.29, and 12.26, respectively. **b, d, f** Survival probability of Amur leopard population at the 100th year under the inbreeding depression scenarios with lethal equivalents of 3.14, 6.29, and 12.26, respectively.

In the case of 12.26 LEs, vaccinating 10 or fewer leopards per year will not prevent the population from going extinct, with the probability of extinction being over 99% in all scenarios (Fig. 3f). The increase in the number of vaccinated individuals per year had little effect on the genetic diversity; the average rates of increase were 1.88% and 1.94% in the cases of 3.14, 6.29, respectively (Fig. 3a, c). In the case of 3.14 LEs, the population achieved growth with more than 8 leopards randomly vaccinated per year, reaching a population size of 90 or more after 100 years (Table S8). Whilst in the cases of 6.29 LEs and 12.26 LEs, vaccination did not change the trend of population decline; in all scenarios, the average final population size was smaller than the initial. In the three inbreeding depression scenarios, the benefits of random vaccination of 4 leopards per year were similar to the benefits of the control of free-ranging domestic dogs (Table S6, Table S8).

**Sensitivity analysis results**. The results of 16 simulations for the sensitivity analysis are presented in Table 1. In the sensitivity analysis of LEs, the lower the inbreeding depression, the lower the risk of population extinction, with the extinction probability decreasing to 2.5% in the scenario without inbreeding depression. Increases in CCI had positive effects on the population size, resulting in a population size of 70 individuals after 100 years when the cycle was 7 years. The population developed well, especially when the mortality after CDV infection (MCI) was reduced by 40%, reaching a population of 151 leopards with almost no risk of extinction in 100 years. The reduction in the mortality of cubs and adults (CMR, FMR, MMR) slowed the population decline; however, the change in mortality of adult males (MMC) had a very limited influence on population size. In

addition, changing the carrying capacity (K) alone also had a limited effect on the population compared to other parameters, and reducing K had a greater effect on the population growth rate than increasing K.

The results of the sensitivity analysis showed that the different output variables had different sensitivities to the parameters, with indices ranging from − 17.04 (largest negative influence) to +14.65 (largest positive influence). The output variables R and N were the most sensitive to BFP, CMR and MCI, and both the decrease in MCI and CMR, and the increase in BFP resulted in large sensitivity indices, both at least +5. The greatest effect on R was found in the increase in BFP (sensitivity index of −17.04). The output variable PE was sensitive to the increase in MCI and CMR and the decrease in BFP, with sensitivity indices of approximately +9. For the output variable GD, the sensitivity to all parameters was low, with the absolute value of the sensitivity index less than 1. For all four output variables, the sensitivity for MMR, and K were very low compared to the other parameters.

## Discussion

We simulated the population dynamics of the leopard population on the Sino-Russian border considering the epidemic of CDV. The extinction probability was greater than 20% for all values of LEs, which is far below the minimum standard for the probability of population survival proposed by Shaffer (1981)[38]. Compared to the baseline scenario, the implementation of management measures improved the population trends. Low-coverage vaccination (6 leopards per year) had the greatest benefit according to the set of assumptions posed, followed by domestic dog control, which had a benefit equivalent to vaccinating 4 leopards per year. The benefits of habitat expansion were small when implemented

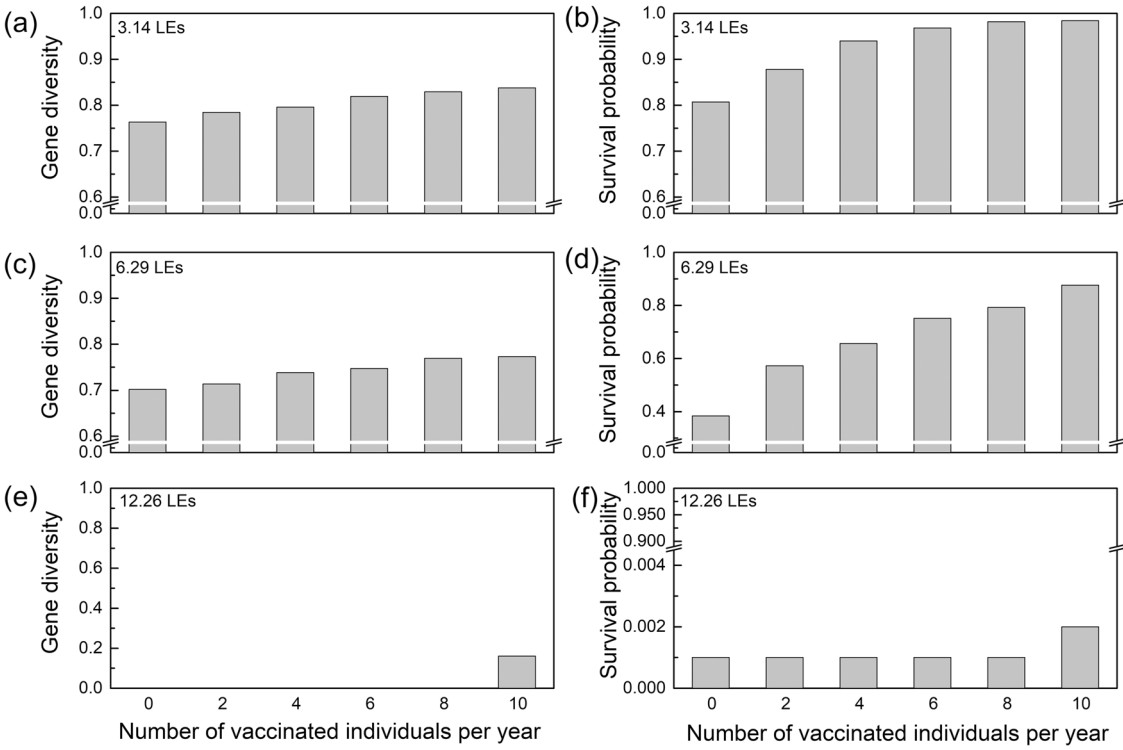

**Fig. 3 Impact of low-coverage vaccination on genetic diversity and survival probability of the Amur leopard population in the Sino–Russian border for three values of inbreeding depression. a, c, e** Gene diversity of Amur leopard population at the 100th year under the inbreeding depression scenarios with lethal equivalents of 3.14, 6.29, and 12.26, respectively. **b, d, f** Survival probability of Amur leopard population at the 100th year under the inbreeding depression scenarios with lethal equivalents of 3.14, 6.29, and 12.26, respectively.

**Table 1 Sensitivity analysis examining the relative influence of 8 parameters in the metamodel. R(SD): mean (standard deviation) stochastic population growth rate; N(SD): mean (standard deviation) number of leopards at year 100; GD(SD): initial gene diversity (heterozygosity) remaining in extant populations at year 100; PE: the probability of extinction, defined as only 1 sex remaining at year 100; $S_R$: mean sensitivity index of the mean stochastic population growth rate; $S_N$: mean sensitivity index of the mean number of leopards; $S_{GD}$: mean sensitivity index of initial gene diversity remaining in extant populations; $S_{PE}$: mean sensitivity index of the probability of extinction.**

| Scenario | R (SD) | N (SD) | GD (SD) | PE | $S_R$ | $S_N$ | $S_{GD}$ | $S_{PE}$ |
|---|---|---|---|---|---|---|---|---|
| Baseline | −0.011 (0.101) | 46(41) | 0.763 (0.131) | 0.202 | | | | |
| Lethal equivalents (LEs) | | | | | | | | |
| 0 | 0.009 (0.079) | 126 (41) | 0.821 (0.1) | 0.025 | +1.761 | +1.746 | +0.075 | −0.876 |
| 6.29 | −0.03 (0.125) | 9 (16) | 0.702 (0.162) | 0.618 | −1.676 | −0.805 | −0.08 | +2.043 |
| Cycles of CDV infection (CCI) | | | | | | | | |
| 3 | −0.023 (0.122) | 20 (28) | 0.701 (0.166) | 0.424 | −2.522 | −1.390 | −0.204 | +2.748 |
| 7 | −0.004 (0.088) | 70 (46) | 0.797 (0.124) | 0.100 | +1.681 | 1.318 | +0.109 | −1.262 |
| Mortality after CDV infection (MCI) | | | | | | | | |
| 40% | −0.057 (0.173) | 0 (1) | 0.475 (0.274) | 0.979 | −10.088 | −2.490 | −0.944 | +9.616 |
| −40% | 0.030 (0.058) | 151 (17) | 0.884 (0.04) | 0.001 | +9.027 | +5.746 | +0.393 | −2.488 |
| Successfully breeding female proportion (BFP) | | | | | | | | |
| 40% | 0.055 (0.055) | 189 (15) | 0.891 (0.04) | 0 | +14.646 | +7.769 | +0.419 | −2.500 |
| −40% | −0.088 (0.176) | 0 (0) | 0 (0) | 0.999 | −17.035 | −2.500 | −2.500 | +9.864 |
| Cubs (0-1 years old) mortality rate (CMR) | | | | | | | | |
| 40% | −0.063 (0.162) | 0 (1) | 0.401 (0.361) | 0.997 | −11.327 | −2.498 | −1.188 | +9.839 |
| −40% | 0.036 (0.062) | 171 (17) | 0.883 (0.041) | 0.001 | +10.553 | +6.817 | +0.393 | −2.488 |
| Adult females (>3 years old) mortality rate (FMR) | | | | | | | | |
| 40% | −0.042 (0.15) | 2 (7) | 0.588 (0.203) | 0.856 | −6.814 | −2.372 | −0.574 | +8.094 |
| −40% | 0.021 (0.064) | 136 (22) | 0.868 (0.051) | 0.005 | +7.102 | +4.88 | +0.341 | −2.438 |
| Adult males (>3 years old) mortality rate (MMR) | | | | | | | | |
| 40% | −0.014 (0.108) | 40 (39) | 0.747 (0.139) | 0.244 | −0.487 | −0.336 | −0.053 | +0.520 |
| −40% | −0.009 (0.097) | 54 (46) | 0.776 (0.143) | 0.165 | +0.465 | +0.457 | +0.041 | −0.458 |
| Carrying capacity (K) | | | | | | | | |
| 40% | −0.010 (0.099) | 62 (59) | 0.777 (0.139) | 0.199 | +0.248 | +0.879 | +0.046 | −0.126 |
| −40% | −0.015 (0.114) | 24 (24) | 0.696 (0.155) | 0.291 | −0.788 | −1.243 | −0.226 | +1.122 |

alone, but in combination with other measures, it would result in further population releases and contribute to the long-term development of the population (Fig. 1).

The strategy for CDV mitigation is to reduce the transmission to leopards by limiting contacts between leopards and domestic dogs or through domestic dog vaccination. Currently, domestic dogs frequently invade Amur leopard habitat[6,39]. On the Chinese side, according to our recent camera-trapping data and field surveys, unvaccinated domestic dogs still enter the parkland and attack wildlife[40]. Domestic dogs are a proven source of CDV for wild animals and they transmit CDV to leopards through direct predation by leopards or indirectly through interaction with other wildlife[41]. Given that CDV is preventable, strict vaccination of domestic dogs is needed. Dog management cannot completely interrupt CDV transmission, as they are not the only source of CDV infection[6]. However, dog management, among all conservation strategies, is the easiest and safest to achieve and, if combined with other measures, it can effectively improve population trajectories. To ensure the long-term success of leopard conservation, we suggest to implement policies aimed at gradually controlling dogs or regularly vaccinating dogs, if not completely prohibiting dogs in the leopard's core range.

For low-coverage vaccination of Amur leopards as a direct way of mitigating CDV, the size, survival probability, and genetic diversity of the population after 100 years increased to varying degrees as the number of individuals vaccinated each year increased. It's already the most profitable among the 3 measures when 6 leopards are randomly vaccinated per year. However, vaccination is not easy to implement for rare and mysterious species such as the leopard; it is often difficult and costly, depending upon the availability of funds and social factors, and requiring enhanced transborder cooperation. The cost of vaccinating tigers was estimated by Gilbert et al. (2020)[6] to be nearly $15,000 per tiger per year; the costs increase linearly with the number of leopards vaccinated per year but the benefit to the population does not increase linearly; thus, there is a need to balance economic costs and population benefits in sustainable long-term conservation. In addition, for regular vaccination for dogs, there is also a large financial investment.

With the establishment of the national park on the Chinese side, the habitat area of the Amur leopard has been expanded, but through observations in the past years, the core distribution area of the Amur leopard remains at the Sino-Russian border and has not spread to China on a large scale. As our simulation results showed, habitat expansion exhibited the worst performance among the three management measures; it barely improved the trend trajectory of the population and yielded essentially a very similar result as the baseline. This was also confirmed in the results of the carrying capacity sensitivity analysis. Conservation actions generally prioritize habitat protection and restoration[42], which seems inconsistent with our results. The reason for this apparent inconsistency may be the fact that the Amur leopard population is facing many threats, such as a small initial population, inbreeding depression, and CDV[15], which have resulted in the population size being maintained below the environmental capacity for long periods. Therefore, changes in carrying capacity may have a relatively weak effect on a population in such conditions. This also suggests that habitat expansion alone could do little to restore the small, isolated population of the leopard, because even if with a sufficiently large habitat, it is difficult for the population to spread due to the high extinction risk. The same result was obtained when assessing the extinction risk of tigers in central India[43].

The results of the three management measures implemented individually revealed that the population decline was only slowed to some extent and the final population decline was not changed.

These findings indicate that it would be difficult to save the Amur leopard by only undertaking unilateral management measures, especially in the case of moderate or higher inbreeding depression, a comprehensive and multifaceted approach is required for population recovery. The best performance of the combination of two measures was the combination of domestic dog control and low-coverage vaccination of leopards, with a survival probability of more than 99% with an inbreeding depression of 3.14 LEs. This indicates that if the population inbreeding depression is not serious, there is potential to ensure the viability of the population and implement the slow recovery of the population with the control of CDV. When the three measures were combined, compared with the combination of two CDV control measures, the expansion of the habitat further increased the population, and the population growth rate and survival probability all increased.

However, since none of the three measures contributed to genetic diversity, all management measures and their combinations were ineffective with high inbreeding depression, especially with 12.26 LEs, for which even the combination of the three measures failed to prevent the extinction of the population. Some studies suggest that the reintroduction or translocation of new individuals may be the best way to relieve the evident inbreeding pressure and genetic drift[44–46]. However, this method is quite controversial and faces challenges, such as the availability of captive-bred individuals and infectious and noninfectious hazards[15,27,47]; it should be deemed a last resort and only considered after the failure of previous conservation strategies[48]. In addition, habitat expansion can also reduce inbreeding depression to some extent[49].

The metamodeling approach requires many uncertain parameters in each model, and the uncertainty will propagate across the models[37]; therefore, a sensitivity analysis was necessary. The sensitivity analysis highlighted that the proportion of successfully breeding females had a primary effect on the probability of population extinction; larger proportions of reproductive females were more conducive to the continuation of the population. The results also showed that female mortality (given the polygynous breeding system) had a greater effect on the population size than male mortality. This is consistent with studies of PVA in fennec foxes (*V. zerda*)[34] and Asian elephants (*Elephas maximus*)[50]. Therefore, it is critical to ensure that female leopards have a good breeding habitat and low human-caused mortality to improve their survival rates.

In addition, the results of the sensitivity analysis showed that the lower the mortality after CDV infection was, the more favorable it was for population growth. This suggests once again that reducing CDV outbreaks within the leopard population is urgently needed. It is worth noting that compared with the Amur tiger, the proportion of the host prey that can carry CDV is higher in the diet of the leopard[51], and this proportion will increase with the decrease in the density of the preferred large and medium-sized ungulate prey[52,53]. The risk of CDV infection increases with the consumption of small carnivorous prey by leopards, coupled with the high virus transmissibility and mortality[41,54]. Each individual Amur leopard is undoubtedly at greater death risk, increasing the extinction risk of this vulnerable and isolated small population. Therefore, for control of CDV in the Amur leopard population, the protection of its preferred ungulate prey is also important. And it is vital to establish strict management policies to prevent poaching of ungulates.

In summary, the conservation of the small Amur leopard population requires a multifaceted effort, including improving the survival environment of females to enhance their reproduction and survival rates, as well as considering the disturbance caused by CDV epidemics. Finally, increasing the management and vaccination of free-roaming domestic dogs in the forest,

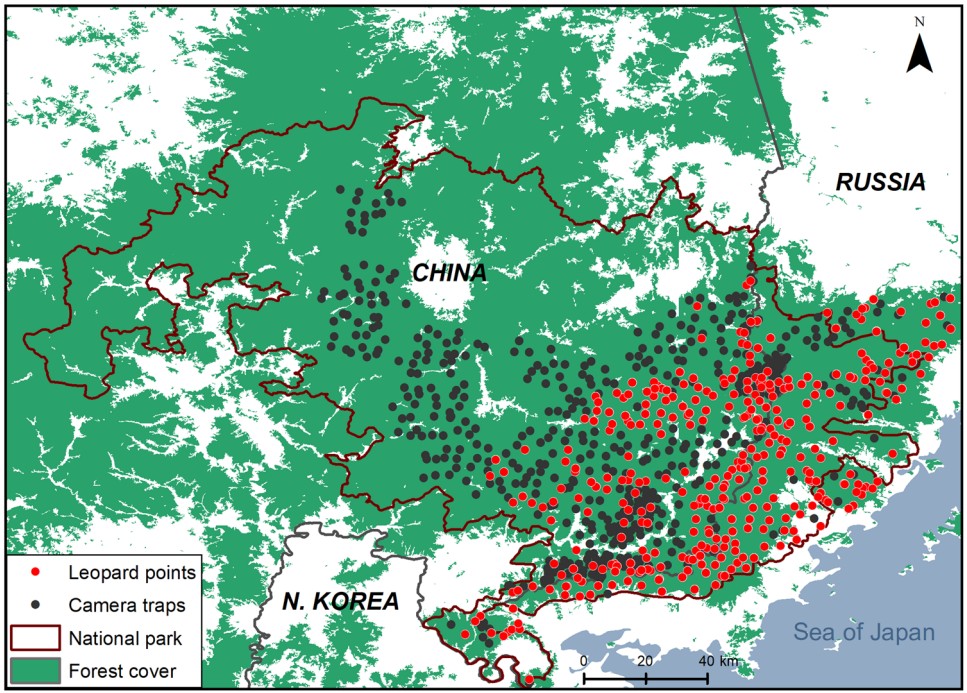

**Fig. 4 The current range of the Amur leopard population across the Sino-Russian border.** According to Vitkalova et al. (2018)[13]. Leopard presence points in the transboundary landscape during 2014-2015 is shown in red dots. General Forest cover are taken from the Advanced Land Observing Satellite (ALOS-2 PALSAR-2, 2014).

vaccinating leopards and gradually expanding the range of the habitat, are all strategies that should be thoroughly considered timely by wildlife. When the inbreeding depression is severe, the synergistic consideration of the various management actions is critical.

We have demonstrated the theoretical and practical application of a metamodel combining the uncertainties of epidemiology and demography in an individual-based context. Although the future trends of the population obtained from our simulation analysis can provide good guidance for management measures to maintain the sustainability of the Amur leopard population, there are challenges in accurately parameterizing the models, thereby affecting to some extent the relative importance of measure, which can also be dependent on model assumptions. Our results should be viewed as demonstrating the relative benefits of various possible measures rather than as absolute, accurate predictions of future population trends. The primary value of the metamodel lies in its ability to integrate and critically analyze available information on the ecology of the Amur leopard population, as well as the ability to use quantitative indicators to assess the resilience of the population. In addition, we did not apply inbreeding depression to adult survival; however, inbreeding could have an impact on the survival and reproduction of adult leopards, especially in the face of changing environmental conditions[55,56]. Therefore, we may have underestimated the effect of inbreeding depression in the PVA.

## Methods

**Study area**. Our study area corresponds to the Sino-Russian forest and has approximately 9,000 km² of Amur leopard core habitat. In a census jointly performed by China and Russia in 2015[13], at least 87 Amur leopards (78 adults and 9 subadults/cubs) were identified in the area[57], based on camera trap monitoring[40] (Fig. 4). This area is characterized by a typical mountainous landscape, and the major vegetation types include secondary deciduous birch (*Betula linn*) and oak (*Quercus mongolica*) forests, as well as some coniferous forests[27]. The main preys of the Amur leopards in this area include sika deer (*Cervus nippon*), wild boars (*Sus scrofa*), roe deer (*Capreolus pygargus*), and free-ranging domestic dogs[51]. The area has been subject to human disturbance for decades, especially on the Chinese side,

including the cultivation of ginseng and other crops, frog farming, grazing, and poaching. Domestic dogs frequently participate in these activities as human companion animals, and cases of domestic dogs being preyed on by tigers and leopards have occurred.

**Modeling overview**. We developed an epidemiological model and a population dynamics model separately and assembled them into a metamodel (Fig. 5). Both models were individual-based and spatially implicit, i.e., they did not explicitly consider spatial positioning of individuals, but implicitly considered spatial interactions in some parameters (e.g., the probability of virus transmission). For the epidemiological model, we used Outbreak (version 2.11.0)[58], for the population dynamics model, we used Vortex (version 10.5.0)[59]. Outbreak integrates only limited aspects of population dynamics[36]; hence, we coupled it Vortex.

We linked the two models through the software package MetaModel Manager (version 1.0.6)[60], which was developed for linking multiple simulation models representing components or processes in an overall system by transferring common data and updated values of variables across interacting simulations. MetaModel Manager handles discrepancies in the time resolution of Outbreak and Vortex: Outbreak has a daily time interval, while Vortex has an annual time interval (see Supplementary Method 1 and 2). We used MetaModel Manager to call an instance of Outbreak to simulate the disease dynamics for a year based on a daily time resolution; then individuals who died due to disease were removed and data were transmitted to Vortex to update yearly population demographics (Fig. 5a). All the demographic factors unrelated to CDV, (e.g., birth, reproductive and death rate) were controlled by Vortex.

**CDV epidemiological model description**. In Outbreak, five states for individuals were defined: pre-susceptible, i.e., the state includes all individuals from birth to the earliest age of susceptibility; susceptible, i.e., capable of becoming infected if exposed to the virus; exposed, i.e., in contact with the virus, but not infected; infectious, i.e., actively shedding the virus and capable of transmitting the disease; recovered, i.e., survived to the infection and assumed to be permanently immune (no longer infectious) (Fig. 5b). The state of each individual in the population was tracked along model runs, and the probabilities of transition among states depended on the number of individuals currently in each state and on other relevant parameters such as specified rates and durations (e.g., incubation), probabilities of encounter, transmission, infection, and recovery[61].

Recovered female leopards pass antibodies to their newborns through lactation, keeping them in pre-susceptibility for the lactation period; other cubs are considered susceptible after birth. Females infected with CDV transmitted the virus to their cubs with a certain probability. Transitions from susceptible to exposed occurred in two ways: one is from the environment, i.e., by predation on domestic dogs and small wild carnivores; the other is the interactions with other infectious leopards. Concerning the transmission from the environment, the prevalence of

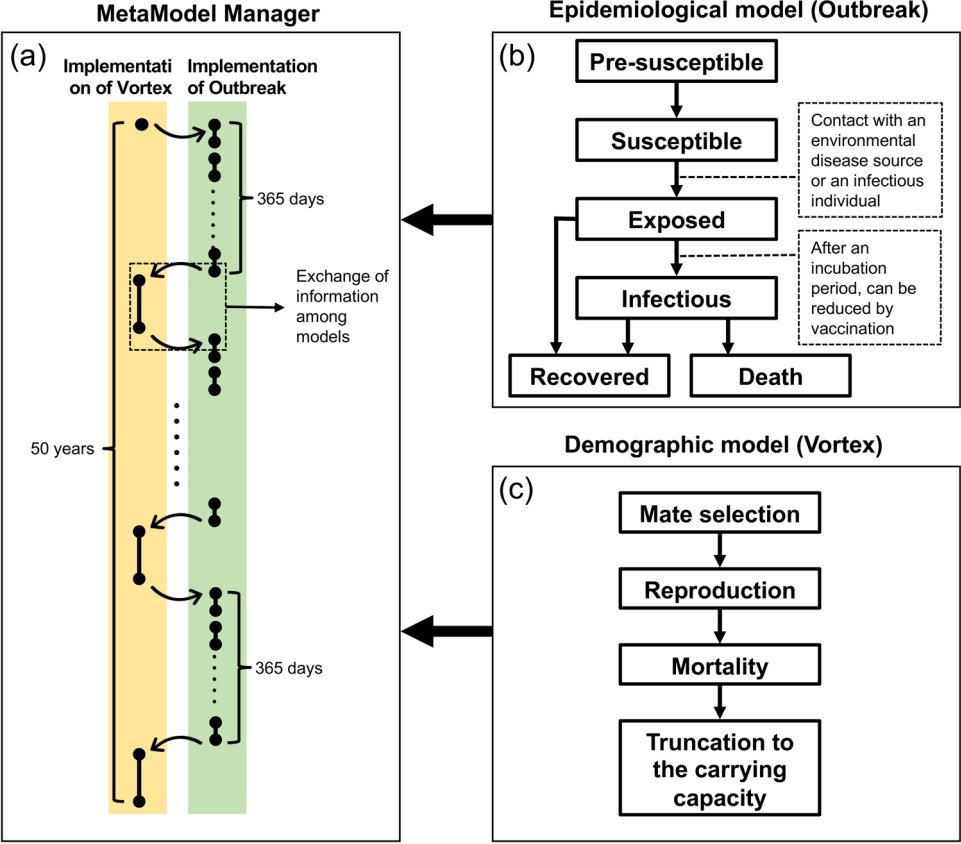

**MetaModel Manager**

**Epidemiological model (Outbreak)**

**Demographic model (Vortex)**

**Fig. 5 Schematic diagram of the interface between Vortex and Outbreak programs as controlled by MetaModel Manager. a** At the beginning of each Vortex time interval, MetaModel Manager retrieves details on the current population, passes this information to Outbreak, and directs Outbreak to simulate the spread of disease within the population over a year MetaModel Manager then retrieves the final disease status of each individual and passes this to Vortex where it is used in the calculation of fertility and mortality rates. **b** Outbreak simulates disease transmission dynamics over a daily time interval, monitoring all individuals in a population as they move between six disease states. **c** The Vortex population model is an individual-based, age- and sex-structured stochastic simulation of the extinction process that operates in this case over an annual time interval. In this model, each year is characterized by a sequence of life history events.

CDV comes in cycles of epizootic waves[62]; we averaged the daily mean probability of virus transmission over a wave and incorporated it in the model as a constant value.

Transitions from the exposed to the infected state occurred with a certain probability after an incubation period and the probability was reduced if the individual is vaccinated. Infectious individuals had the potential to transmit the virus to other individuals during a certain infectious period, after which infectious individuals either recovered (acquiring permanent immunity) with a certain probability or died.

All the parameter values of the baseline epidemiological model are given in Table 2. While some parameters had a unique value, other parameters were assigned a uniformly distributed random value ranging between two extremes. Here we provide the explanation for the choice of the value for some parameters, while Supplementary Method 1 provides a complete explanation. When calculating the probability of transmission of the virus from the environment, we postulated that leopards were primarily infected with CDV by preying on small- and medium-sized carnivores that were carrying the virus. Therefore, for calculating this probability, we estimated the number of small carnivores preyed per year. From previous studies[63,64], we estimated the annual biomass of prey requirement of females with cubs (considering a 30% of biomass utilization loss): 1,998.81 kg. The number of domestic dogs and each small carnivore species consumed per year per leopard was calculated following Sugimoto et al. (2016)[51] (Table S1). The probability of each small carnivore species carrying the virus was based on epizootic cycles: there are years of high and of low infection rates (Table S1). The average annual probability and the daily mean probability of CDV infection from the environment for Amur leopards in a susceptible state during an epizootic cycle (in the baseline model, we set the epizootic cycle of CDV (CCI) to 5 years with a low infection risk as a background and a high infection risk year every five years) was 0.288 and 0.00079, respectively.

Since Amur leopards are solitary animals, interacting only during the mating period, we assumed that the average effective interaction among leopards does not depend on the population size and occurs 1-2 times per month, resulting in a daily individual encounter probability of 0.05. The probability of transmitting the virus

in an encounter was estimated to be the same as the current annual mean prevalence of CDV from the environment[41]. These estimates are conservative as no studies have provided data on the possibility of transmitting CDV through scent markers.

**Demographic model description.** Vortex models population dynamics as a sequence of discrete events for each individual based on user-defined probability distributions. At the beginning of each year, the ages of individuals were increased by one unit, and relevant information was passed from the Outbreak model. All individuals were assumed to have the longest lifespan and a first-birth age. The carrying capacity corresponded to the maximum number of leopards that can stay in the area. We considered the effect of inbreeding depression, which is the decrease in survival fitness with increased genome-wide homozygosity that occurs in the offspring of related parents[65]. We quantified the severity of inbreeding depression in terms of lethal equivalents (LEs). One lethal equivalent corresponds to a group of deleterious alleles that would cause one death on average if made homozygous[66]. Vortex simulates inbreeding depression in two ways: one is through recessive lethal alleles, in which the lethal alleles can be removed from the population through the mortality of previous generations of inbreeding; on the second way, selection is ineffective at purging inbreeding depression when inbreeding depression results from a general advantage of heterozygotes over all homozygotes (or to a lesser extent, when it is caused by recessive sublethal alleles). We set a value for the percent of the inbreeding depression due to recessive lethal alleles.

The series of simulated events consisted of mate selection, reproduction, mortality, and truncation to the carrying capacity (Fig. 5c). Concerning mate selection, the Amur leopard is polygamous. In the model, only individuals reaching sexual maturity were able to find a mate (with a certain proportion of individuals breeding) and they no longer mate after a certain age. Concerning reproduction, females had a certain maximum number of broods per year and the number of cubs in each brood followed an assigned probability distribution. Newborns were assigned a sex according to a given sex ratio. Cubs depended on their mother for a certain period, during which their mother could carry out mating and

**Table 2 List of parameters with baseline values for the canine distemper epidemiological model for the Amur leopard population on the Sino-Russian border in Outbreak.**

| State | Parameter | Baseline value | Unit |
|---|---|---|---|
| Pre-susceptible | Probability that an individual never becomes susceptible | 0 | - |
| | Transmission probability from an infectious mother to a newborn | 1 | - |
| | Time that maternally derived immunity protects an offspring | 150-180 | days |
| Susceptible | Daily transmission probability from the environment | 0.00078904 | - |
| | Average number of other individuals encountered by an individual per day | 0.05 | individuals |
| | Transmission probability among leopard individuals during an encounter | 0.288 | - |
| Exposed | Duration of the incubation period | 2-7 | days |
| Infectious | Duration of the infectious period | 30-60 | days |
| Recovered | Probability of recovering and acquiring permanent immunity | 0.6 | - |
| | Mortality rate after CDV infection (MCI) | 0.4 | - |

For some parameters (where the baseline value is expressed as a range), the value is imputed randomly from a uniform distribution within the two extremes. The parameter assigned an acronym is used in the sensitivity analysis.

**Table 3 List of parameters and their baseline simulation values for the demographic model for the Amur leopard population on the Sino-Russian border in Vortex.**

| Parameter | Baseline value | Unit |
|---|---|---|
| Longest lifespan | 12 | year |
| Maximum age of reproduction | 12 | year |
| Age of first birth (female/male) | 3/3 | year |
| Successfully breeding female proportion (BFP) | 65 | % |
| Breeding male proportion | 70 | % |
| Maximum number of broods per year | 1 | individual |
| Percentage of litters of different sizes | | |
| 1 offspring | 30 | % |
| 2 offspring | 60 | % |
| 3 offspring | 10 | % |
| Share of males at birth | 0.50 | - |
| Time of cubs' dependency on mother | 1 | year |
| Cubs (0–1 years old) mortality rate (CMR) | 40 | % |
| Subadult females (1–3 years old) mortality rate | 11.8 | % |
| Subadult males (1–3 years old) mortality rate | 19.5 | % |
| Adult females (>3 years old) mortality rate (FMR) | 8.2 | % |
| Adult males (>3 years old) mortality rate (MMR) | 8.2 | % |
| Carrying capacity (K) | 56 (adult females) | individuals |
| Lethal equivalents (LEs) | 3.14 | - |
| Percent due to recessive lethal alleles | 50 | % |

The parameters assigned an acronym or a symbol are used in the sensitivity analysis.

reproduction. If the female leopard died, all the dependent offspring also died. Mortality was modeled as a random event with age-dependent probabilities. Concerning the truncation to the carrying capacity, after the population was updated, a number of leopards corresponding to the quantity exceeding the carrying capacity were randomly removed.

All the parameter values of the baseline demographic model are given in Table 3. In the natural environment, the duration of pregnancy of the Amur leopard is 90–95 days, and the lactation period is 4-6 months. The cubs will not leave their mother until at least one year of age. Due to the lack of mortality data for the Amur leopard, we referred to the data for the African leopard (*P. pardus*) from 2002 to 2007 in the Mkhuze Game Reserve, South Africa[67], where the distractions such as hunting are similar to those in the Sino-Russian border area.

The carrying capacity was calculated based on the home range, defined as the area in which an animal lives and moves. The home range overlap is high for adult male leopards but relatively low for females[68] and the home range of males overlaps with that of several females; cubs and juveniles do not maintain their home range. Therefore, in our study, the carrying capacity was defined as the number of female individuals accommodated. According to the study results of

Rozhnov et al. (2015)[69] using the minimum convex polygon method (MCP, 95% utilization probability), a habitat area of 9,000 km$^2$ can accommodate 56 female leopards with a home range of 160 km$^2$. We set the LEs of the baseline model to 3.14 based on a study of inbreeding in juveniles from 40 populations[70]. The detailed software description and all demographic parameters used in the Vortex model are described in detail in Supplementary Method 2.

**Scenarios**. We defined a scenario as a simulation of the metamodel with specific initial conditions, parameter configurations and output trajectories, representative of a particular situation or experiment. In a scenario, some parameter values were randomly sampled from an assigned probability distribution, and some events were stochastic (e.g., the transition of an individual among epidemiological states). Therefore, following a Monte Carlo approach[71], simulations consisted of 1000 model runs, each one with a specific draw of parameter values from its probability distributions; the results of population size and population genetic diversity were averaged across the runs. The population survival probability was calculated as the percentage simulations ending with population survival. The baseline scenario corresponded to the currently observed situation without any management intervention. To avoid underestimation of the degree of inbreeding depression in wildlife populations[72,73], we tested 6.29 and 12.26 LEs from the O'Grady et al. (2006)[73] meta-analysis, in addition to the test of inbreeding depression of 3.14 LEs, to assess population trends under different cases of inbreeding depression. In the simulation of 6.29 LEs, 3.94 LEs were used to impact fecundity, and 2.35 LEs to impact first-year survival. In the simulation of 12.26 LEs, we added 5.97 LEs for altering survival from age 1 to sexual maturity based on 6.29 LEs.

We developed two types of scenarios: management alternatives assessment and sensitivity analysis. The purpose of the management alternatives assessment was to evaluate the effect of different management measures and combinations thereof on the future development of the Amur leopard population. The purpose of the sensitivity analysis was to evaluate the sensitivity of the response of the metamodel to parameter changes and to identify the parameters in the model that had a strong influence on population development. All scenarios were defined over a 100-year time horizon and were initialized with 87 individuals based on Feng et al. (2017)[57].

**Management alternatives assessment**. We considered three management alternatives: (A) controlling the domestic dogs in the Amur leopard habitat, i.e., taking protective measures to prevent leopards from contracting CDV from predation of domestic dogs; (B) low-coverage vaccination of leopards; (C) habitat expansion. We set up scenarios or families of scenarios with parametric configurations or input trajectories representing the implementation of these management practices, or combinations thereof. For management alternative (A), we eliminated CDV transmission from domestic dogs to Amur leopards and recalculated the transmission probability, which reduced the mean daily probability of infection from the environment to 0.00066 (scenario A). For management alternative (B), we considered the vaccination of 6 leopards per year; a single vaccination was assumed to induce lifelong protective immunity with a 0.9 probability (scenario B). For management alternative (C), we set the habitat area in the Chinese part of the study area to spread to 8,600 km$^2$ as estimated by Jiang et al. (2015)[28] which increased the carrying capacity to 73 female leopards in total (scenario C). After 20 years, the environmental carrying capacity remained unaltered. Concerning management alternative (B), to analyze the influence of the number of individuals vaccinated per year, we set up a family of scenarios corresponding to different vaccination intensities (2, 4, 6, 8, and 10 leopards vaccinated each year). Alternative (A) and (B) represent measures directed towards the control of CDV, while for alternative (C), although habitat expansion can also reduce the transmission of CDV to some extent, in this study, habitat expansion is aimed at increasing the caring capacity for the population, not at mitigating the effects of CDV.

**Sensitivity analysis**. For eight chosen key parameters, we ran two scenarios with two test values, one higher and one lower than the baseline, keeping the other parameters constant. The parameters were chosen to reflect different aspects of population and disease dynamics, and some were based on parameters explored in previous studies[37,41,74]: LEs, CCI, mortality after CDV infection (MCI), breeding female proportion (BFP), cubs (0-1 year old) mortality rate (CMR), adult (>3 years old) females mortality rate (FMR), adult (>3 years) male mortality rate (MMR), carrying capacity (K). We evaluated the sensitivity of model output variables $Y$ to a parameter $P$ with the formula: $S_Y = ((Y_1 - Y_0)/|Y_0|)/((P_1 - P_0)/|P_0|)$ where $Y_0$ and $P_0$ correspond to the value of mean model output variable and the parameter tested in the baseline scenario, and $Y_1$, $P_1$ correspond to the values of mean model output variable and the test value for the parameter in the modified scenario. A positive $S_Y$ indicated that the test value had a positive effect on the output, and a negative $S_Y$ meant that the test value had a negative effect on this output.

A total of 16 simulations were carried out: with test values for LEs of 0 and 6.29, CCI of 3 and 7 years and for the other 6 parameters, values plus or minus 40% of the baseline values were tested[50], and this range of variation covers the possible parameter changes in real situations. The model outputs observed in the analysis were sensitivity index ($S_Y$), mean stochastic population growth rate (R), mean population size at the 100th year (N), population genetic diversity at the 100th year (GD), probability of extinction (PE).

**Reporting summary**. Further information on research design is available in the Nature Research Reporting Summary linked to this article.

## Data availability
Main data generated or analysed during this study are included in this published article (and its supplementary information files). More detailed simulation data is available in Science Data Bank at https://doi.org/10.57760/sciencedb.02762.

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

## Acknowledgements

This work was supported by the National Natural Science Foundation of China (31971539), the National Science and Technology Basic Resources Survey Program of China (2019FY101700) and a scholarship from the China Scholarship Council (202106040062).

## Author contributions

T.W. and D.W. conceived the ideas and designed the study; D.W. involved in the development of the PVA model, writing–original draft, and visualization. F.A. provided guidance on model logic and draft writing. J.L.D.S. advised on the analyses. All authors contributed critically to the drafts, and gave final approval for publication.

## Competing interests

The authors declare no competing interests.
