## [Peer Review File · Communications Biology]

Reviewers' comments:

Reviewer #1 (Remarks to the Author):

This valuable paper integrates two modeling approaches (an epidemiological model constructed using OUTBREAK, and a demographic model constructed using VORTEX) to produce a population viability model for the critically endangered Amur leopard population across the Sino-Russian border. The model considers the impact of infection with canine distemper virus (which has recently been recognized in this population) and inbreeding depression as an important threat to a population numbering ~80 individuals. The authors use their model to assess the comparative impact of three management strategies, vaccination of domestic dogs, low coverage vaccination of leopards and expansion of available habitat.

Overall, the paper is a well executed and much-needed contribution that will be of value for the management of both this and other small populations of threatened species. The paper is unusual in addressing questions related to the practical management disease threats to endangered populations. However, I feel that the authors may have placed too much weight in their conclusions as to the relative benefits of their three interventions (dog vaccination > leopard vaccination > habitat improvement). The relative importance of each of these management interventions is highly sensitive to the assumptions made when parameterizing the model in ways that I will explain below. Nevertheless, the findings of this paper are still extremely valuable and with some acknowledgement of the challenges in accurately parameterizing models and the effects this can have on conclusions a revised manuscript should be a very valuable addition to the literature.

The demographic and epidemiological parameters seem credible although it should be noted that apparently minor overestimates or underestimates of these parameters can have considerable consequences when modelled over 50 years.

The baseline scenario seems unrealistically negative. Of course, models that describe such complex systems can never hope to be a perfect predictor of real world ecology it is important that they resemble them as closely as is practical. In reality the Amur leopard population has been constrained into an isolated habitat island in Southwest Primorye and has expanded into all available habitat. With the improvements in habitat protection and quality on the Chinese side of the border the population has begun to expand across the border (as indicated in Figure 1). This would suggest a population which is exhibiting net growth, which is not reflected in the baseline scenarios which suggests a population in decline. This is presumably due to either an underestimate of reproductive potential, or an overestimate of mortality (either related to CDV, inbreeding, demography or a combination). One consequence of this might be to underestimate the potential contribution of Management Scenario C, the expansion of available habitat as the modelled population is pre-programmed not to be able to expand into available habitat. This is important, as it is possible that for a population that is growing towards carrying capacity habitat expansion might be the most important management strategy and far outperform either CDV control strategy which leaves the reader with a completely different conclusion (i.e. dog vaccination might not be the most effective management strategy).

The conclusion that control of CDV in free-roaming dogs was the most effective management strategy was interesting, particularly as research on the Russian side of the border has concluded that this approach would be ineffective, focussing instead on low coverage vaccination of big cats (tigers in that case, see Gilbert et al 2020 PNAS 117(50): 31954-31962). The findings of the current study are surprising, considering the relatively low contribution of dogs as a source of infection (as shown by the daily transmission probability of 0.00084 from all carnivores reducing to only 0.00066 under management scenario 3 when dogs are excluded). Although only three dogs are eaten per year (compared to 18 wild carnivores, Table S1), the estimate of CDV prevalence used is higher than for all other species (1.5% to 5.0%). Estimating prevalence is fraught with difficulties and the authors have chosen reasonable values based on the published literature. However, it should be noted that

estimates based on field data from dogs in Primorye give a CDV prevalence of 0.77% (Gilbert et al 2020), which is at least half the prevalence used in the current study. Considering that this estimate was based on serology results in a population where seroprevalence varied from 15.9 - 41.4%, it is likely that the actual CDV prevalence among dogs on the Chinese side of the border is similar (as the authors report seroprevalence of 37%).

Taken together, I wonder how conclusions would be impacted if scenarios were to be re-run with a baseline population where the leopard population were increasing, and where dog prevalence were reduced. It may or may not be justified to include such analyses in a revised manuscript, but they should at least be assessed in the response to the review to determine whether the conclusions currently drawn in the paper still hold.

A few other minor comments were made while reviewing the manuscript and are detailed below:

Introduction:

Lines 69-70: [CDV is...] "...regarded as the main threat to the conservation of large felids worldwide...". I am assuming that the authors are referring to CDV being the pathogen of greatest threat to large felids worldwide, not (as written) as the main threat of any kind (of which, large felids face many that are of greater importance than pathogens).

Lines 72-73: "...Bengal tigers (*P. t. tigris*) (Mulia et al. 2021)...". Mulia et al 2021 refers to Sumatran tigers (*P. t. sumatrae*), not Bengal.

Lines 81-83: "For the Amur leopard, because the management of wild animals poses challenges, the focus was on the control of domestic dogs (Gilbert et al. 83 2015; Goodrich et al. 2011)". Neither of these sources address CDV in Amur leopards, nor do they endorse management strategies focused on dogs (indeed the Goodrich reference doesn't refer to dogs at all).

Line 100: Minor typo - "...there is no research to available to a..."

Lines 127-129: Minor typo - "Our analysis results will provide a inform the management policy for the Amur leopard's persistence."

Method:

Line 135: Should be "individual cubs".

Lines 177-178: "All individuals were considered in pre-susceptible state from birth and a proportion of them became susceptible after a certain lifetime period." - The term 'pre-susceptible' isn't clearly defined. Cubs born to immune dams might be considered immune until the waning of maternal antibodies. But it isn't clear why cubs of susceptible dams wouldn't be susceptible themselves, or at what stage they would move from the pre-susceptible to susceptible classifications. Can you clarify?

Line 213: I'd think it was more typical to reserve the term "rutting" for describing the mating season of certain ruminants. Breeding or mating period is probably better in this context.

Discussion:

Lines 442-444: "Dog management cannot completely interrupt CDV transmission, as they are the only source of CDV infection (Gilbert et al. 2020)" ...surely this should read "Dog management cannot completely interrupt CDV transmission, as they are NOT the only source of CDV infection (Gilbert et al. 2020)"?

Lines 468-471: Although the cost of vaccinating leopards is high, it should be emphasized that all management scenarios are expensive. Elimination of CDV in the free-ranging dog population would require that the proportion of the population requiring vaccination is $1 - (1/R_0)$ [where R_0 is the

effective reproductive value of CDV]. Estimating R_0 is challenging, but for example to eliminate a pathogen with an R_0 of 5.0 would require $1 - (1/5) = 0.8$ of the population to be vaccinated. The numbers of dogs in the vicinity of leopard habitat is likely to number in the range of X0,000s or X00,000s. Delivering vaccine (2 doses initially, with boosters every 3 years) to 80% of those dogs on an indefinite basis is far from inexpensive and is likely to exceed $N \times \$15,000$ (where N = the number of leopards vaccinated each year).

Lines 529-533: I am in complete agreement with the authors that while models should not be used as predictors of reality they are still a useful tool in comparing the impact of various intervention strategies and the importance of unknown values. However, care needs to be taken in drawing conclusions of the comparative importance of different management strategies as these are highly sensitive to assumptions made when selecting parameters (as detailed in my comments on the relative importance of the three management strategies).

Reviewer #2 (Remarks to the Author):

Manuscript (ID: COMMSBIO-22-1266-T) on Contributions of distemper control and habitat expansion to the Amur leopard viability

The paper has important conservation value in policy level decision making and it is written relatively well with appropriate analysis. However, I have minor suggestions given below.

Abstract, Line 35

Change "small-isolated threated population" to small-isolated threatened population

Introduction,

In introduction there is no rationale explained that how habitat expansion can influence impact of CDV. Also, the manuscript highlights the dogs as potential carrier but fails to explain the action/ mode of transmission of infection from dog to wild animals. Overall reason for initiation of this study can be strengthened.

Line 97, Don't know how appropriate to use land conditions which this context. And what are the land conditions can be explained further. May be reworded

line 100 Remove "to" from there is no research to available

Line 105 -108, Here it has been mentioned that Vortex model is very comprehensive but authors have shows only inbreeding depression as component of the model. To assert its comprehensiveness all important component of the model should be highlighted.

Line 128-129, Not clear. Rewrite the sentence for clarity.

Materials and Methods

Line 136, or cubs individuals – how many?

Line 140, Change free-range domestic dogs to free-ranging domestic dogs

The abbreviation (LEs) given to Lethal equivalents and Inbreeding depression. Please recheck and correct it.

Line 213-216, Before mentioning this, effect of population size and encounter rate can be highlighted with citations.

Results and discussion

Overall done well

How realistic is your viability analysis metamodel? Need more details about its application for revival of threatened wildlife population in discussion which is more practical application oriented.

Line 463-466, It is better to give probable reason for inconsistency in the model.

Line 493-494, Other potential strategies to mitigate inbreeding can be considered mentioning in this paragraph

Line 508-509, Measures to ensure protection can also be highlighted

The manuscript should explain/suggest the way forward and follow up actions required based on your core findings. How your findings are going to help in minimising further decline of species.

In Table 1 and Table 2, you need to give references for all your baseline values. You can create another column and add the references.

Reviewer #3 (Remarks to the Author):

Population viability analysis (PVA) is a general tool for assessing viability of the endangered population. The author used PVA to assess the effectiveness of different management conservation measures on the critically endangered Amur leopard population under various inbreeding depression scenarios. This study can be a tool for making decisions with imperfect knowledge of the system state of wildlife population, especially on limited timelines.

Although the PVA requires numerous parameters and long term monitoring work, which increases the uncertainty of the model to some extent, this study gave detailed parameter estimates and sources of the required parameters in the supporting information, which made the model reproducible. In terms of method, the author used a novel PVA metamodel incorporating CDV epidemiology to provide practice guidelines for Amur leopard conservation, which has important insights for small populations of endangered species.

However, the paper has the following shortcomings, should be improved:

(1) In the sensitivity analysis, for the 6 parameters, such as Mortality after CDV infection (MCI), values plus or minus 40% of the baseline values were tested. Why 40% was chosen instead of other values, please explain.

(2) In the methods section, especially the model description (CDV epidemiological model description, Demographic model description), takes up too much space and could be considered to be shortened or moved partly to the Supporting Information.

(3) The dog control measure is optimal when the three conservation measures are implemented individually, and this should be clearly stated in the discussion and possibly do some work to test it in next work.

(4) Note the use of italics in the main document, e.g. in line 327.

(5) The paragraph in line 433, and the paragraph in line 463 are both CDV control measures, but the paragraph in line 449 is a habitat expansion measure, so it is suggested to exchange the position of the paragraphs in lines 463 and 449 to ensure the coherence of the content.

August 4, 2022

Dear Reviewers,

We are very grateful for the detailed feedback that you three anonymous reviewers provided on our *Communications Biology* submission: “**Contributions of distemper control and habitat expansion to the Amur leopard viability**” (Manuscript ID: COMMSBIO-22-1266-T). We think that the feedback resulted in valuable additions, changes, and improvements to our manuscript.

We have carefully thought about the reviewers' comments, moreover, we have attempted to address each of these comments in the current revision of the paper. We attract the attention to the following point: based on reviewers' comments, we have revised a model parameter and corrected an error in the previous version of the model so that the model results are no longer unrealistically negative. We feel that the paper has been considerably strengthened.

We have provided a point-by-point response to the comments made by reviewers. Comments are numbered, so that they can be easily identified and cross-referenced among them. In each case, we begin by restating each of the comments and immediately below each comment, we indicate how we addressed each issue and if applicable, where the changes occur within the manuscript. Having said this, we remain open to any additional suggestions that you might subsequently offer. Additionally, should it be that we either did not fully address a point or misinterpreted a point we remain more than willing to make subsequent alterations to the manuscript.

Once again, please accept our thanks for providing us with the opportunity to resubmit our manuscript to *Communications Biology*.

Sincerely,
Tianming Wang

Responses to Reviewer 1

Thank you for your constructive comments for our paper! We believe that we have been able to address your comments and suggestions, and that our paper has been substantially improved as a result. Especially based on your comments, we found a mistake in a parameter of the model, which was very important for this study. Of course, the mistake is now addressed, and all the parameters of the model are now well checked. Below, we indicate how we responded to each of your comments.

[R1C1] This valuable paper integrates two modeling approaches (an epidemiological model constructed using OUTBREAK, and a demographic model constructed using VORTEX) to produce a population viability model for the critically endangered Amur leopard population across the Sino-Russian border. The model considers the impact of infection with canine distemper virus (which has recently been recognized in this population) and inbreeding depression as an important threat to a population numbering ~80 individuals. The authors use their model to assess the comparative impact of three management strategies, vaccination of domestic dogs, low coverage vaccination of leopards and expansion of available habitat. Overall, the paper is a well executed and much-needed contribution that will be of value for the management of both this and other small populations of threatened species. The paper is unusual in addressing questions related to the practical management disease threats to endangered populations.

Thank you very much for your kind words about the content of the paper and the model used.

[R1C2] However, I feel that the authors may have placed too much weight in their conclusions as to the relative benefits of their three interventions (dog vaccination > leopard vaccination > habitat improvement). The relative importance of each of these management interventions is highly sensitive to the assumptions made when parameterizing the model in ways that I will explain below. Nevertheless, the findings of this paper are still extremely valuable and with some acknowledgement of the challenges in accurately parameterizing models and the effects this can have on conclusions a revised manuscript should be a very valuable addition to the literature.

We agree. Based on your comments below, we have modified one of the parameters of the CDV (see R1C4 for details) and found a mistake in the parameter setting of in the model (see R1C3 for details). After addressing these points, we find that the relative benefits of the different measures change, and we really should not be overly concerned with the ranking of the relative benefits among the different measures. So based on this suggestion, we have made changes to the abstract and discussion and

other relevant content. The relative benefits of the three interventions are no longer emphasized, as they depend on the assumptions and parameterization. As in the Discussion, we have deleted “Domestic dog control was significantly better than low-coverage vaccination and habitat expansion or even their combination” and rewritten it as:

“Low-coverage vaccination (6 leopards per year) had the greatest benefit according to the set of assumptions posed, followed by domestic dog control, which had a benefit equivalent to vaccinating 4 leopards per year. The benefits of habitat expansion were small when implemented alone, but in combination with other measures, it would result in further population releases and contribute to the long-term development of the population”. (lines 419-424)

We reiterated the challenge of model parameter estimation in section 4.3:

“Although the future trends of the population obtained from our simulation analysis can provide good guidance for management measures to maintain the sustainability of the Amur leopard population, there are challenges in accurately parameterizing the models, thereby affecting to some extent the relative importance of measure, which can also be dependent on model assumptions. Our results should be viewed as demonstrating the relative benefits of various possible measures rather than as absolute, accurate predictions of future population trends.” (lines 527-533)

[R1C3] The demographic and epidemiological parameters seem credible although it should be noted that apparently minor overestimates or underestimates of these parameters can have considerable consequences when modelled over 50 years. The baseline scenario seems unrealistically negative. Of course, models that describe such complex systems can never hope to be a perfect predictor of real world ecology it is important that they resemble them as closely as is practical. In reality the Amur leopard population has been constrained into an isolated habitat island in Southwest Primorye and has expanded into all available habitat. With the improvements in habitat protection and quality on the Chinese side of the border the population has begun to expand across the border (as indicated in Figure 1). This would suggest a population which is exhibiting net growth, which is not reflected in the baseline scenarios which suggests a population in decline. This is presumably due to either an underestimate of reproductive potential, or an overestimate of mortality (either related to CDV, inbreeding, demography or a combination). One consequence of this might be to underestimate the potential contribution of Management Scenario C, the expansion of available habitat as the modelled population is pre-programmed not to be able to expand into available habitat. This is important, as it is possible that for a population that is growing towards

carrying capacity habitat expansion might be the most important management strategy and far outperform either CDV control strategy which leaves the reader with a completely different conclusion (i.e. dog vaccination might not be the most effective management strategy).

Thank you very much for these detailed comments. Through your suggestions, we did a deep investigation in the model parameterization and we found that there was indeed a mistake in the previous parameter configuration, which led to negative results. In exploring why the model results were unrealistically negative we found that when we linked the two submodels (Vortex and Outbreak), the simulation of CDV in the Outbreak model did not successfully replace the simulation of CDV infection that we had previously set up during the phase of model testing. This resulted in the previous version of the manuscript with simulation results that amplified the negative impact of the CDV, leading to all simulations being negative. We are sorry that we did not catch this mistake before submission the manuscript, and sincerely thank you for finding the simulation results negative and prompting us to find the mistake. After careful and repeated checks, we have ensured that similar mistakes do not occur again in the model. In addition to this, we may have overestimated the risk of CDV infection in dogs based on your comments, which also led to negative results (see R1C4 for details), and we have made changes based on your comments.

Based on the above two modifications, the model results are no longer unrealistically negative. The 50-year simulations are not sufficient to compare the differences between the scenarios, especially with respect to the survival probability, which after 50 years remains above 90% for all scenarios under the 3.14 and 6.29 LEs inbreeding depression, and it is difficult to distinguish which protection measures are better. We have therefore extended the simulation period from 50 to 100 years to make it easier to compare the simulation results between scenarios. The Table 3, Figure 3, Figure 4, Figure 5 are modified to include this change

Habitat expansion is indeed very important for endangered species, but through our observations, although the establishment of the national park on the Chinese side has expanded the habitat area of the Amur leopard, the core range of the leopard is still at the Sino-Russian border and has not spread to the interior of China on a large scale (refer to Figure 1 in the main text for details). The high mortality risk caused by factors such as CDV, inbreeding depression and lack of preys limits the spread of leopards. So, we wrote in the text: “The reason for this apparent inconsistency may be the fact that the Amur leopard population is facing many threats, such as a small initial population, inbreeding depression, and CDV, which have resulted in the population size being maintained below the environmental capacity for long periods.” (see lines 461-464). Our re-simulation showed that under a 3.14 LEs inbreeding depression scenario, habitat expansion alone maintained the population size at 88 individuals at 50th years, which basically maintained the initial population size, but at 100th years, the population size dropped to 57 individuals, showing that habitat

expansion can maintain the population size in the short term, but does not guarantee the long-term population persistence. In the scenario where CDV control is overlaid with habitat expansion measures, the population could reach 139 individuals at 100th years, so our revised paper emphasized the importance of a combination of measures (refer to R1C2).

To make the above information clearer, we add the following to the discussion about habitat expansion:

“With the establishment of the national park on the Chinese side, the habitat area of the Amur leopard has been expanded, but through observations in the past years, the core distribution area of the Amur leopard remains at the Sino-Russian border and has not spread to China on a large scale. As our simulation results showed, habitat expansion exhibited the worst performance among the three management measures;” (lines 453-457)

[R1C4] The conclusion that control of CDV in free-roaming dogs was the most effective management strategy was interesting, particularly as research on the Russian side of the border has concluded that this approach would be ineffective, focussing instead on low coverage vaccination of big cats (tigers in that case, see Gilbert et al 2020 PNAS 117(50): 31954-31962). The findings of the current study are surprising, considering the relatively low contribution of dogs as a source of infection (as shown by the daily transmission probability of 0.00084 from all carnivores reducing to only 0.00066 under management scenario 3 when dogs are excluded). Although only three dogs are eaten per year (compared to 18 wild carnivores, Table S1), the estimate of CDV prevalence used is higher than for all other species (1.5% to 5.0%). Estimating prevalence is fraught with difficulties and the authors have chosen reasonable values based on the published literature. However, it should be noted that estimates based on field data from dogs in Primorye give a CDV prevalence of 0.77% (Gilbert et al 2020), which is at least half the prevalence used in the current study. Considering that this estimate was based on serology results in a population where seroprevalence varied from 15.9 - 41.4%, it is likely that the actual CDV prevalence among dogs on the Chinese side of the border is similar (as the authors report seroprevalence of 37%). Taken together, I wonder how conclusions would be impacted if scenarios were to be re-run with a baseline population where the leopard population were increasing, and where dog prevalence were reduced. It may or may not be justified to include such analyses in a revised manuscript, but they should at least be assessed in the response to the review to determine whether the conclusions currently drawn in the paper still hold.

Based on your suggestions, we used field data estimates for dogs in Primorye with a

CDV prevalence of 0.77% (Gilbert et al., 2020), which is closer to the real data in study area. We re-simulated all scenarios after modifying this parameter and fixing the mistake in the Vortex (see R1C3 for details), and the simulation results were no longer so overly negative that we needed to extend the simulation time to distinguish between the results of the different scenarios.

The conclusions about the paper have changed somewhat from the previous dog control measures being the most effective to vaccination being the most effective (6 per year), but based on your comments that different measures have different values, we made more prudential statements about the relative effectiveness of the three interventions (see R1C2 for details).

[R1C5] Lines 69-70: [CDV is...] “...regarded as the main threat to the conservation of large felids worldwide...”. I am assuming that the authors are referring to CDV being the pathogen of greatest threat to large felids worldwide, not (as written) as the main threat of any kind (of which, large felids face many that are of greater importance than pathogens).

Thank you very much for your comments. Of course, we agree and we have revised this sentence to

“regarded as the pathogen of greatest threat to large felids worldwide (Terio et. al., 2013).” (line 63)

[R1C6] Lines 72-73: “...Bengal tigers (*P. t. tigris*) (Mulia et al. 2021)...”. Mulia et al 2021 refers to Sumatran tigers (*P. t. sumatrae*), not Bengal.

Sorry for this mistake, we have modified it to “Sumatran tigers (*P. t. sumatrae*) (Mulia et al. 2021)” (see lines 66)

[R1C7] Lines 81-83: “For the Amur leopard, because the management of wild animals poses challenges, the focus was on the control of domestic dogs (Gilbert et al. 83 2015; Goodrich et al. 2011).”. Neither of these sources address CDV in Amur leopards, nor do they endorse management strategies focused on dogs (indeed the Goodrich reference doesn’t refer to dogs at all).

We agree, but we think that the conclusion made about tigers by Gilbert et al. (2015) can support our opinion, with opportune specifications.

We removed the reference to Goodrich et al. 2011 and moved Gilbert et al. 2015 to the previous sentence and made some changes:

“the main strategy for the prevention and control of CDV in endangered species is the control

of the infection pool (as concluded for tigers by Gilbert et al. (2015)).” (lines 72-73)

[R1C8] Line 100: Minor typo - “...there is no research to available to a...”

We modified it to

“there is no research available to assess the benefits of this measure for the Amur leopard population.” (lines 90-91).

Thanks.

[R1C9] Lines 127-129: Minor typo - “Our analysis results will provide a inform the management policy for the Amur leopard's persistence.”

We modified it to

“The results of our analysis can inform the management policy for the Amur leopard's persistence.” (lines 116-117).

Thanks.

[R1C10] Line 135: Should be “individual cubs”.

Sorry for the confusion, we modified it to

“78 adults and 9 subadults/cubs” (line 122)

(refer to R2C8)

[R1C11] Lines 177-178: “All individuals were considered in pre-susceptible state from birth and a proportion of them became susceptible after a certain lifetime period.” - The term ‘pre-susceptible’ isn’t clearly defined. Cubs born to immune dams might be considered immune until the waning of maternal antibodies. But it isn’t clear why cubs of susceptible dams wouldn’t be susceptible themselves, or at what stage they would move from the pre-susceptible to susceptible classifications. Can you clarify?

Thanks for bringing our attention to this point of ambiguity. According to the Outbreak manual, the pre-susceptible state includes all individuals from birth to the earliest age of susceptibility, so we made sure that this is clear in the new version of the manuscript.

“pre-susceptible, *i.e.*, the state includes all individuals from birth to the earliest age of

susceptibility” (lines 153-154)

In fact, there is no evidence that newborn Amur leopard cubs are not susceptible to CDV. Therefore, cubs are susceptible after birth and may become infected within the first couple days. In the pre-susceptible category, we set to zero the probability that an individual never becomes susceptible. However, pre-susceptibility occurs in case cubs are born to mothers with immunity as they retain antibodies for 150–180 days. The above parameter settings are given in Table 1.

For better clarity, we directly replace “All individuals were considered in pre-susceptible state from birth and a proportion of them became susceptible after a certain lifetime period.” With:

“Recovered female leopards pass antibodies to their newborns through lactation, keeping them in pre-susceptibility for the lactation period; other cubs are considered susceptible after birth.” (lines 163-165).

[R1C12] Line 213: I’d think it was more typical to reserve the term “rutting” for describing the mating season of certain ruminants. Breeding or mating period is probably better in this context.

Thank you for your comment. We changed it to "mating period."

[R1C13] Lines 442-444: “Dog management cannot completely interrupt CDV transmission, as they are the only source of CDV infection (Gilbert et al. 2020)” ...surely this should read “Dog management cannot completely interrupt CDV transmission, as they are NOT the only source of CDV infection (Gilbert et al. 2020)”?

Yes, this does miss the "not". Thank you for catching the error.

[R1C14] Lines 468-471: Although the cost of vaccinating leopards is high, it should be emphasized that all management scenarios are expensive. Elimination of CDV in the free-ranging dog population would require that the proportion of the population requiring vaccination is $1-(1/R_0)$ [where R_0 is the effective reproductive value of CDV]. Estimating R_0 is challenging, but for example to eliminate a pathogen with an R_0 of 5.0 would require $1-(1/5) = 0.8$ of the population to be vaccinated. The numbers of dogs in the vicinity of leopard habitat is likely to number in the range of X0,000s or X00,000s. Delivering vaccine (2 doses initially, with boosters every 3 years) to 80% of those dogs on an indefinite basis is far from inexpensive and is likely to exceed $N \times \$15,000$ (where N = the number of leopards vaccinated each year).

It is true that dog control is also costly. In fact, what we are trying to emphasize in this sentence is not the difference in costs between the different measures, but rather the fact that the number of leopards vaccinated should be taken into account when developing a vaccination program. As the number of leopards vaccinated increases, the costs increase linearly but the benefits do not, so it is important to balance the benefits and the economic costs. To make this clearer, we have added a sentence:

“The costs increase linearly with the number of leopards vaccinated per year but the benefit to the population does not increase linearly”. (lines 448-450)

Nevertheless, based on your comments, we have added:

“In addition, for regular vaccination for dogs, there is also a large financial investment.” (lines 451-452)

[R1C15] Lines 529-533: I am in complete agreement with the authors that while models should not be used as predictors of reality they are still a useful tool in comparing the impact of various intervention strategies and the importance of unknown values. However, care needs to be taken in drawing conclusions of the comparative importance of different management strategies as these are highly sensitive to assumptions made when selecting parameters (as detailed in my comments on the relative importance of the three management strategies).

Thank you very much for your suggestion and we fully agree with you. In the revised manuscript we no longer highlight the relative benefits of different measures, but instead call for a combination of different measures which are more effective than single measure implementation. In the discussion we added, to acknowledge the challenge of parameter selection:

“there are challenges in accurately parameterizing the models, thereby affecting to some extent the relative importance of measure, which can also be dependent on model assumptions” (see lines 529-531)

Responses to Reviewer 2

Thank you for your constructive comments for our paper. After making modifications based on your comments, we have greatly improved the details of the manuscript. Below, we indicate how we responded to each of your comments.

[R2C1] The paper has important conservation value in policy level decision making and it is written relatively well with appropriate analysis.

Thank you for your kind words about the manuscript and your encouragement for our work.

Abstract

[R2C2] Line 35, Change “small-isolated threatened population” to small-isolated threatened population

Agreed. We have modified it according to your comment.

Introduction

[R2C3] There is no rationale explained that how habitat expansion can influence impact of CDV. Also, the manuscript highlights the dogs as potential carrier but fails to explain the action/ mode of transmission of infection from dog to wild animals. Overall reason for initiation of this study can be strengthened.

This paper focuses on modeling the effects of CDV control (in two ways: dog control and leopard vaccination) and habitat expansion on leopard populations. Habitat expansion is a separate measure that aims to increase the carrying capacity for the population and is not intended to slow the effects of CDV, however, for sake of scenario comparison, we prefer to include it, as it gives to the reader important information about the limiting factors affecting the leopard population.

To make it clearer, we have added the following

“Alternative (A) and (B) represent measures directed towards the control of CDV, while for alternative (C), although habitat expansion can also indirectly reduce the transmission of CDV to some extent, in this study, habitat expansion is aimed at increasing the carrying capacity for the population, not at mitigating the effects of CDV.” (line 290-293)

The transmission of CDV from dogs to leopards is mainly through leopard predation on dogs, but of course, in addition to dogs, leopards can also be infected with CDV through predation on other small carnivores, as we explain:

“The leopard can prey on various virus hosts (free-ranging domestic dogs and small sized carnivores such as the Asian badger *Meles meles*, red fox *Vulpes vulpes*, and leopard cat

Prionailurus bengalensis) that act as infection pools.” (see lines 69-71)

This is why dog control alone cannot completely eliminate the spread of CDV in leopard populations, which is why we have carried out a simulated CDV vaccination of leopards.

In addition, we explain the two ways of CDV transmission to leopards:

“Transitions from susceptible to exposed occurred in two ways: one is from the environment, *i.e.*, by predation on domestic dogs and small wild carnivores; the other is the interactions with other infectious leopards” (lines 166-168)

[R2C4] Line 97, Don't know how appropriate to use land conditions which this context. And what are the land conditions can be explained further. May be reworded

Sorry for the ambiguity. We refer to the habitat in terms of area, therefore we changed the “land conditions” to “area conditions”

[R2C5] line 100 Remove “to” from there is no research to available

Agreed. We have modified it, thank you for finding this mistake.

[R2C6] Line 105 -108, Here it has been mentioned that Vortex model is very comprehensive but authors have shown only inbreeding depression as component of the model. To assert its comprehensiveness all important component of the model should be highlighted.

Vortex incorporates density constraints, genetic factors, spatial dynamics and other functions in addition to inbreeding depression, but not all of these functions were applied in this study, for example, density constraints and spatial dynamics were not considered in this study. The sentence to which this comment is referred is intended to emphasize the importance of the inbreeding depression function contained in Vortex for this study. So, to avoid confusion, based on your suggestion, we have revised this paragraph to:

“Vortex software package has a wide range of applications in PVA, which has been used to evaluate the survival status and different management strategies for the conservation of endangered species, such as the griffon vulture (*Gyps fulvus*), fennec fox (*V. zerda*), and mountain lion (*Puma concolor*) (Aresu et al. 2021; Benson et al. 2019; Franklin et al. 2021). In addition, Vortex incorporates inbreeding depression (the organism's reduced ability to survive as a result of the inbreeding of related individuals), which is very important for small-sized populations like the object of our study.” (lines 94-100)

[R2C7] Line 128-129, Not clear. Rewrite the sentence for clarity.

Thank you for your comments. We rewrote this way:

“The results of our analysis can inform the management policy for the Amur leopard's persistence.” (see lines 116-117)

[R2C8] In Materials and Methods, line 136, or cubs individuals – how many?

The nine individuals included subadults and cubs. To express more clearly, we modified it to “78 adults and 9 subadults/cubs” (line 122) (refer to R1C10)

[R2C9] Line 140, Change free-range domestic dogs to free-ranging domestic dogs

Thank you for your comment, we changed all “free-range” to “free-ranging” in the new version of the manuscript.

[R2C10] The abbreviation (LEs) given to Lethal equivalents and Inbreeding depression. Please recheck and correct it.

We gave the abbreviation (LEs) to Lethal Equivalents, and we modified the Inbreeding depression in Table 2 to Lethal equivalents. Thanks.

[R2C11] Line 213-216, Before mentioning this, effect of population size and encounter rate can be highlighted with citations.

Since Amur leopards are solitary animals. After our discussion, we feel that the effect of changes in population size on the transmission of CDV in Amur leopard population is limited, and this effect is not considered in this study. And the increase in encounter rate certainly accelerates the transmission of CDV in the leopard population, which is common sense, so we did not put extra emphasis on it, but made a reasonable estimation of the encounter rate based on the life habits of Amur leopards.

Discussion

[R2C12] How realistic is your viability analysis metamodel? Need more details about its application for revival of threatened wildlife population in discussion which is more practical application oriented.

Of course, the metamodel is not predictive of the real world, as pointed out by Reviewer 1 in R1C2 and R1C15. The results are oriented and serve to give a prediction in terms of relative importance of different management strategies. The important is that limits are acknowledged and results are discussed respecting the assumptions made.

Also like Reviewer 3 said in R3C1: “This study can be a tool for making decisions with imperfect knowledge of the system state of wildlife population, especially on limited timelines.” And we can explore more deeply in the next work like the Reviewer 3 suggested in R3C4

To make this point more clearly, we have adjusted section 4.3 of the discussion as follows:

“We have demonstrated the theoretical and practical application of a metamodel combining the uncertainties of epidemiology and demography in an individual-based context. Although the future trends of the population obtained from our simulation analysis can provide good guidance for management measures to maintain the sustainability of the Amur leopard population, there are challenges in accurately parameterizing the models, thereby affecting to some extent the relative importance of measure, which can also be dependent on model assumptions. Our results should be viewed as demonstrating the relative benefits of various possible measures rather than as absolute, accurate predictions of future population trends.”
(lines 525-533)

[R2C13] Line 463-466, It is better to give probable reason for inconsistency in the model.

Since the results have changed a bit due to rerunning the model (see R1C3 and R1C4 for details), the inconsistency is now absent and, consequently, we removed the description.

[R2C14] Line 493-494, Other potential strategies to mitigate inbreeding can be considered mentioning in this paragraph

Thanks for the suggestion, we agree.

We added, at the end of the paragraph.

“In addition, habitat expansion can also reduce inbreeding depression to some extent.” (see lines 491-492)

[R2C15] Line 508-509, Measures to ensure protection can also be highlighted

Agreed, we added, at the end of the paragraph:

“And it is vital to establish strict management policies to prevent poaching of ungulates.”
(see lines 514-515)

[R2C16] The manuscript should explain/suggest the way forward and follow

up actions required based on your core findings. How your findings are going to help in minimizing further decline of species.

Thanks for this suggestion:

We wrote:

“In summary, the conservation of the small Amur leopard population requires a multifaceted effort, including improving the survival environment of females to enhance their reproduction and survival rates, as well as considering the disturbance caused by CDV epidemics. Finally, increasing the management and vaccination of free-roaming domestic dogs in the forest, vaccinating leopards and gradually expanding the range of the habitat, are all strategies that should be thoroughly considered timely by wildlife. When the inbreeding depression is severe, the synergistic consideration of the various management actions is critical.” (see lines 516-523)

[R2C17] In Table 1 and Table 2, you need to give references for all your baseline values. You can create another column and add the references.

Thank you very much for your comments. We understand that this would be practical. However, most of the parameters used are not directly cited from other articles, but are obtained by secondary calculations using data provided by other articles. For this reason, it is quite complicated to fill the table in a synthetic way. Respectfully, we prefer to use the text and the Supplementary Information to explain the detailed process of calculations, and the references used. In addition, some parameters were reasonably estimated and did not have references, such as the probability that an individual never becomes susceptible, the transmission probability from an infectious mother to a newborn, the time that maternally derived immunity protects an offspring. The value of these parameters are based on reasonable estimation from the knowledge of the species life-history traits.

Responses to Reviewer 3

Thank you for your comments concerning our manuscript. Your comments are all valuable and very helpful for revising and improving our paper, as well as the important guiding significance to our research. Below, we indicate how we responded to each of your comments.

[R3C1] Population viability analysis (PVA) is a general tool for assessing viability of the endangered population. The author used PVA to assess the effectiveness of different management conservation measures on the critically endangered Amur leopard population under various inbreeding depression scenarios. This study can be a tool for making decisions with imperfect knowledge of the system state of wildlife population, especially on limited timelines. Although the PVA requires numerous parameters and long term monitoring work, which increases the uncertainty of the model to some extent, this study gave detailed parameter estimates and sources of the required parameters in the Supplementary Information, which made the model reproducible. In terms of method, the author used a novel PVA metamodel incorporating CDV epidemiology to provide practice guidelines for Amur leopard conservation, which has important insights for small populations of endangered species.

Thank you very much for your recognition and encouragement for our research

[R3C2] In the sensitivity analysis, for the 6 parameters, such as Mortality after CDV infection (MCI), values plus or minus 40% of the baseline values were tested. Why 40% was chosen instead of other values, please explain.

Thanks for this comment, we agree that this might be somehow arbitrary. We chose $\pm 40\%$ for two main reasons: (1) the $\pm 40\%$ range of variation has been used in published articles (He et al. 2020), and we have added this reference in the text. (2) we think this range of variation covers the possible parameter changes in real situations.

In the text we have added the following:

“A total of 16 simulations were carried out: with test values for LEs of 0 and 6.29, CCI of 3 and 7 years and for the other 6 parameters, values plus or minus 40% of the baseline values were tested⁵⁸, and this range of variation covers the possible parameter changes in real situations.” (lines 320-322)

There are indeed challenges regarding the selection of parameters, and we have added a relevant discussion:

“Although the future trends of the population obtained from our simulation analysis can provide good guidance for management measures to maintain the sustainability of the Amur leopard population, there are challenges in accurately parameterizing the models, thereby affecting to some extent the relative importance of measure, which can also be dependent on model assumptions” (line 585-589)

Please, also consider that the metric used as sensitivity index is normalized by the parameter change. This should limit the arbitrariness of the choice of the parameter.

[R3C3] In the methods section, especially the model description (CDV epidemiological model description, Demographic model description), takes up too much space and could be considered to be shortened or moved partly to the Supplementary Information.

Thank you very much for your comments. Respectfully, we prefer to keep the description of the methods as is. Before submitting the first version of the manuscript, we had already made choices and some detail is in the Supplementary Information. And we believe that the information provided in the text is necessary for a complete high-level description of the models. However, we have tried to shorten this section as much as possible. Starting from 1668 words, arriving to 1250 words, reduced by 418 words.

[R3C4] The dog control measure is optimal when the three conservation measures are implemented individually, and this should be clearly stated in the discussion and possibly do some work to test it in next work.

Since we modified the parameters on the probability of CDV infection in dogs and corrected errors in the model, in the new results, dog control obtains approximately the same result as the benefit of vaccinating 4 leopards per year, while the benefit of vaccinating 6 leopards per year is better than dog control, so the comparison of relative benefits between measures here depends on the number of leopards vaccinated per year. So, we do not overemphasize the relative benefits of the measures (please refer to [R1C2]). Also thank you very much for providing us with research ideas for the next step of exploring in more detail the impact of domestic dogs on wildlife.

[R3C5] Note the use of italics in the main document, e.g. in line 327.

Thanks, we have corrected it and we have been consistent throughout the new version of the manuscript.

[R3C6] The paragraph in line 433, and the paragraph in line 463 are both CDV control measures, but the paragraph in line 449 is a habitat expansion

measure, so it is suggested to exchange the position of the paragraphs in lines 463 and 449 to ensure the coherence of the content.

Thank you for your valuable comments. We have adjusted the paragraphs to make the content of the article coherent.

REVIEWERS' COMMENTS:

Reviewer #2 (Remarks to the Author):

Well Done. I have no further comments.

Reviewer #3 (Remarks to the Author):

It's quality is improved by this revision greatly.

September 21, 2022

Dear Reviewers,

We are very pleased that we got the approval of the anonymous reviewers for the revised manuscript “**Contributions of distemper control and habitat expansion to the Amur leopard viability**” (Manuscript ID: COMMSBIO-22-1266-A) submitted to *Communications Biology*.

Since the reviewers did not provide suggestions for further content modification (see below for comments), this revision was mainly based on the formatting adjustments required by *Communications Biology*.

REVIEWERS' COMMENTS:

Reviewer #2:

Well Done. I have no further comments.

Reviewer #3:

Its quality is improved by this revision greatly.

Thanks again to the three reviewers for their previous valuable comments on the manuscript, which were crucial to its publication.

The following sections of the document are our previous point-by-point responses to reviewers' comments for your reference.

Sincerely,
Tianming Wang

Previous point-by-point responses to reviewers' comments

Responses to Reviewer 1

Thank you for your constructive comments for our paper! We believe that we have been able to address your comments and suggestions, and that our paper has been substantially improved as a result. Especially based on your comments, we found a mistake in a parameter of the model, which was very important for this study. Of course, the mistake is now addressed, and all the parameters of the model are now well checked. Below, we indicate how we responded to each of your comments.

[R1C1] This valuable paper integrates two modeling approaches (an epidemiological model constructed using OUTBREAK, and a demographic model constructed using VORTEX) to produce a population viability model for the critically endangered Amur leopard population across the Sino-Russian border. The model considers the impact of infection with canine distemper virus (which has recently been recognized in this population) and inbreeding depression as an important threat to a population numbering ~80 individuals. The authors use their model to assess the comparative impact of three management strategies, vaccination of domestic dogs, low coverage vaccination of leopards and expansion of available habitat. Overall, the paper is a well executed and much-needed contribution that will be of value for the management of both this and other small populations of threatened species. The paper is unusual in addressing questions related to the practical management disease threats to endangered populations.

Thank you very much for your kind words about the content of the paper and the model used.

[R1C2] However, I feel that the authors may have placed too much weight in their conclusions as to the relative benefits of their three interventions (dog vaccination > leopard vaccination > habitat improvement). The relative importance of each of these management interventions is highly sensitive to the assumptions made when parameterizing the model in ways that I will explain below. Nevertheless, the findings of this paper are still extremely valuable and with some acknowledgement of the challenges in accurately parameterizing models and the effects this can have on conclusions a revised manuscript should be a very valuable addition to the literature.

We agree. Based on your comments below, we have modified one of the parameters of the CDV (see R1C4 for details) and found a mistake in the parameter setting of in the model (see R1C3 for details). After addressing these points, we find that the relative benefits of the different measures change, and we really should not be overly

concerned with the ranking of the relative benefits among the different measures. So based on this suggestion, we have made changes to the abstract and discussion and other relevant content. The relative benefits of the three interventions are no longer emphasized, as they depend on the assumptions and parameterization. As in the Discussion, we have deleted “Domestic dog control was significantly better than low-coverage vaccination and habitat expansion or even their combination” and rewritten it as:

“Low-coverage vaccination (6 leopards per year) had the greatest benefit according to the set of assumptions posed, followed by domestic dog control, which had a benefit equivalent to vaccinating 4 leopards per year. The benefits of habitat expansion were small when implemented alone, but in combination with other measures, it would result in further population releases and contribute to the long-term development of the population”. (lines 419-424)

We reiterated the challenge of model parameter estimation in section 4.3:

“Although the future trends of the population obtained from our simulation analysis can provide good guidance for management measures to maintain the sustainability of the Amur leopard population, there are challenges in accurately parameterizing the models, thereby affecting to some extent the relative importance of measure, which can also be dependent on model assumptions. Our results should be viewed as demonstrating the relative benefits of various possible measures rather than as absolute, accurate predictions of future population trends.” (lines 527-533)

[R1C3] The demographic and epidemiological parameters seem credible although it should be noted that apparently minor overestimates or underestimates of these parameters can have considerable consequences when modelled over 50 years. The baseline scenario seems unrealistically negative. Of course, models that describe such complex systems can never hope to be a perfect predictor of real world ecology it is important that they resemble them as closely as is practical. In reality the Amur leopard population has been constrained into an isolated habitat island in Southwest Primorye and has expanded into all available habitat. With the improvements in habitat protection and quality on the Chinese side of the border the population has begun to expand across the border (as indicated in Figure 1). This would suggest a population which is exhibiting net growth, which is not reflected in the baseline scenarios which suggests a population in decline. This is presumably due to either an underestimate of reproductive potential, or an overestimate of mortality (either related to CDV, inbreeding, demography or a combination). One consequence of this might be to underestimate the potential contribution of Management Scenario C, the expansion of available habitat as the modelled population is pre-

programmed not to be able to expand into available habitat. This is important, as it is possible that for a population that is growing towards carrying capacity habitat expansion might be the most important management strategy and far outperform either CDV control strategy which leaves the reader with a completely different conclusion (i.e. dog vaccination might not be the most effective management strategy).

Thank you very much for these detailed comments. Through your suggestions, we did a deep investigation in the model parameterization and we found that there was indeed a mistake in the previous parameter configuration, which led to negative results. In exploring why the model results were unrealistically negative we found that when we linked the two submodels (Vortex and Outbreak), the simulation of CDV in the Outbreak model did not successfully replace the simulation of CDV infection that we had previously set up during the phase of model testing. This resulted in the previous version of the manuscript with simulation results that amplified the negative impact of the CDV, leading to all simulations being negative. We are sorry that we did not catch this mistake before submission the manuscript, and sincerely thank you for finding the simulation results negative and prompting us to find the mistake. After careful and repeated checks, we have ensured that similar mistakes do not occur again in the model. In addition to this, we may have overestimated the risk of CDV infection in dogs based on your comments, which also led to negative results (see R1C4 for details), and we have made changes based on your comments.

Based on the above two modifications, the model results are no longer unrealistically negative. The 50-year simulations are not sufficient to compare the differences between the scenarios, especially with respect to the survival probability, which after 50 years remains above 90% for all scenarios under the 3.14 and 6.29 LEs inbreeding depression, and it is difficult to distinguish which protection measures are better. We have therefore extended the simulation period from 50 to 100 years to make it easier to compare the simulation results between scenarios. The Table 3, Figure 3, Figure 4, Figure 5 are modified to include this change

Habitat expansion is indeed very important for endangered species, but through our observations, although the establishment of the national park on the Chinese side has expanded the habitat area of the Amur leopard, the core range of the leopard is still at the Sino-Russian border and has not spread to the interior of China on a large scale (refer to Figure 1 in the main text for details). The high mortality risk caused by factors such as CDV, inbreeding depression and lack of preys limits the spread of leopards. So, we wrote in the text: “The reason for this apparent inconsistency may be the fact that the Amur leopard population is facing many threats, such as a small initial population, inbreeding depression, and CDV, which have resulted in the population size being maintained below the environmental capacity for long periods.” (see lines 461-464). Our re-simulation showed that under a 3.14 LEs inbreeding depression scenario, habitat expansion alone maintained the population size at 88

individuals at 50th years, which basically maintained the initial population size, but at 100th years, the population size dropped to 57 individuals, showing that habitat expansion can maintain the population size in the short term, but does not guarantee the long-term population persistence. In the scenario where CDV control is overlaid with habitat expansion measures, the population could reach 139 individuals at 100th years, so our revised paper emphasized the importance of a combination of measures (refer to R1C2).

To make the above information clearer, we add the following to the discussion about habitat expansion:

“With the establishment of the national park on the Chinese side, the habitat area of the Amur leopard has been expanded, but through observations in the past years, the core distribution area of the Amur leopard remains at the Sino-Russian border and has not spread to China on a large scale. As our simulation results showed, habitat expansion exhibited the worst performance among the three management measures;” (lines 453-457)

[R1C4] The conclusion that control of CDV in free-roaming dogs was the most effective management strategy was interesting, particularly as research on the Russian side of the border has concluded that this approach would be ineffective, focussing instead on low coverage vaccination of big cats (tigers in that case, see Gilbert et al 2020 PNAS 117(50): 31954-31962). The findings of the current study are surprising, considering the relatively low contribution of dogs as a source of infection (as shown by the daily transmission probability of 0.00084 from all carnivores reducing to only 0.00066 under management scenario 3 when dogs are excluded). Although only three dogs are eaten per year (compared to 18 wild carnivores, Table S1), the estimate of CDV prevalence used is higher than for all other species (1.5% to 5.0%). Estimating prevalence is fraught with difficulties and the authors have chosen reasonable values based on the published literature. However, it should be noted that estimates based on field data from dogs in Primorye give a CDV prevalence of 0.77% (Gilbert et al 2020), which is at least half the prevalence used in the current study. Considering that this estimate was based on serology results in a population where seroprevalence varied from 15.9 - 41.4%, it is likely that the actual CDV prevalence among dogs on the Chinese side of the border is similar (as the authors report seroprevalence of 37%). Taken together, I wonder how conclusions would be impacted if scenarios were to be re-run with a baseline population where the leopard population were increasing, and where dog prevalence were reduced. It may or may not be justified to include such analyses in a revised manuscript, but they should at least be assessed in the response to the review to determine whether the conclusions currently drawn in the paper still hold.

Based on your suggestions, we used field data estimates for dogs in Primorye with a CDV prevalence of 0.77% (Gilbert et al., 2020), which is closer to the real data in study area. We re-simulated all scenarios after modifying this parameter and fixing the mistake in the Vortex (see R1C3 for details), and the simulation results were no longer so overly negative that we needed to extend the simulation time to distinguish between the results of the different scenarios.

The conclusions about the paper have changed somewhat from the previous dog control measures being the most effective to vaccination being the most effective (6 per year), but based on your comments that different measures have different values, we made more prudential statements about the relative effectiveness of the three interventions (see R1C2 for details).

[R1C5] Lines 69-70: [CDV is...] “...regarded as the main threat to the conservation of large felids worldwide...”. I am assuming that the authors are referring to CDV being the pathogen of greatest threat to large felids worldwide, not (as written) as the main threat of any kind (of which, large felids face many that are of greater importance than pathogens).

Thank you very much for your comments. Of course, we agree and we have revised this sentence to

“regarded as the pathogen of greatest threat to large felids worldwide (Terio et. al., 2013).”
(line 63)

[R1C6] Lines 72-73: “...Bengal tigers (*P. t. tigris*) (Mulia et al. 2021)...”. Mulia et al 2021 refers to Sumatran tigers (*P. t. sumatrae*), not Bengal.

Sorry for this mistake, we have modified it to “Sumatran tigers (*P. t. sumatrae*) (Mulia et al. 2021)” (see lines 66)

[R1C7] Lines 81-83: “For the Amur leopard, because the management of wild animals poses challenges, the focus was on the control of domestic dogs (Gilbert et al. 83 2015; Goodrich et al. 2011).”. Neither of these sources address CDV in Amur leopards, nor do they endorse management strategies focused on dogs (indeed the Goodrich reference doesn’t refer to dogs at all).

We agree, but we think that the conclusion made about tigers by Gilbert et al. (2015) can support our opinion, with opportune specifications.

We removed the reference to Goodrich et al. 2011 and moved Gilbert et al. 2015 to the previous sentence and made some changes:

“the main strategy for the prevention and control of CDV in endangered species is the control of the infection pool (as concluded for tigers by Gilbert et al. (2015)).” (lines 72-73)

[R1C8] Line 100: Minor typo - “...there is no research to available to a...”

We modified it to

“there is no research available to assess the benefits of this measure for the Amur leopard population.” (lines 90-91).

Thanks.

[R1C9] Lines 127-129: Minor typo - “Our analysis results will provide a inform the management policy for the Amur leopard's persistence.”

We modified it to

“The results of our analysis can inform the management policy for the Amur leopard's persistence.” (lines 116-117).

Thanks.

[R1C10] Line 135: Should be “individual cubs”.

Sorry for the confusion, we modified it to

“78 adults and 9 subadults/cubs” (line 122)

(refer to R2C8)

[R1C11] Lines 177-178: “All individuals were considered in pre-susceptible state from birth and a proportion of them became susceptible after a certain lifetime period.” - The term ‘pre-susceptible’ isn’t clearly defined. Cubs born to immune dams might be considered immune until the waning of maternal antibodies. But it isn’t clear why cubs of susceptible dams wouldn’t be susceptible themselves, or at what stage they would move from the pre-susceptible to susceptible classifications. Can you clarify?

Thanks for bringing our attention to this point of ambiguity. According to the Outbreak manual, the pre-susceptible state includes all individuals from birth to the earliest age of susceptibility, so we made sure that this is clear in the new version of the manuscript.

“pre-susceptible, *i.e.*, the state includes all individuals from birth to the earliest age of susceptibility” (lines 153-154)

In fact, there is no evidence that newborn Amur leopard cubs are not susceptible to CDV. Therefore, cubs are susceptible after birth and may become infected within the first couple days. In the pre-susceptible category, we set to zero the probability that an individual never becomes susceptible. However, pre-susceptibility occurs in case cubs are born to mothers with immunity as they retain antibodies for 150–180 days. The above parameter settings are given in Table 1.

For better clarity, we directly replace “All individuals were considered in pre-susceptible state from birth and a proportion of them became susceptible after a certain lifetime period.” With:

“Recovered female leopards pass antibodies to their newborns through lactation, keeping them in pre-susceptibility for the lactation period; other cubs are considered susceptible after birth.” (lines 163-165).

[R1C12] Line 213: I’d think it was more typical to reserve the term “rutting” for describing the mating season of certain ruminants. Breeding or mating period is probably better in this context.

Thank you for your comment. We changed it to "mating period."

[R1C13] Lines 442-444: “Dog management cannot completely interrupt CDV transmission, as they are the only source of CDV infection (Gilbert et al. 2020)” ...surely this should read “Dog management cannot completely interrupt CDV transmission, as they are NOT the only source of CDV infection (Gilbert et al. 2020)”?

Yes, this does miss the "not". Thank you for catching the error.

[R1C14] Lines 468-471: Although the cost of vaccinating leopards is high, it should be emphasized that all management scenarios are expensive. Elimination of CDV in the free-ranging dog population would require that the proportion of the population requiring vaccination is $1-(1/R_0)$ [where R_0 is the effective reproductive value of CDV]. Estimating R_0 is challenging, but for example to eliminate a pathogen with an R_0 of 5.0 would require $1-(1/5) = 0.8$ of the population to be vaccinated. The numbers of dogs in the vicinity of leopard habitat is likely to number in the range of X0,000s or X00,000s. Delivering vaccine (2 doses initially, with boosters every 3 years) to 80% of those dogs on an indefinite basis is far from inexpensive and is likely to exceed $N \times \$15,000$ (where $N =$

the number of leopards vaccinated each year).

It is true that dog control is also costly. In fact, what we are trying to emphasize in this sentence is not the difference in costs between the different measures, but rather the fact that the number of leopards vaccinated should be taken into account when developing a vaccination program. As the number of leopards vaccinated increases, the costs increase linearly but the benefits do not, so it is important to balance the benefits and the economic costs. To make this clearer, we have added a sentence:

“The costs increase linearly with the number of leopards vaccinated per year but the benefit to the population does not increase linearly”. (lines 448-450)

Nevertheless, based on your comments, we have added:

“In addition, for regular vaccination for dogs, there is also a large financial investment.” (lines 451-452)

[R1C15] Lines 529-533: I am in complete agreement with the authors that while models should not be used as predictors of reality they are still a useful tool in comparing the impact of various intervention strategies and the importance of unknown values. However, care needs to be taken in drawing conclusions of the comparative importance of different management strategies as these are highly sensitive to assumptions made when selecting parameters (as detailed in my comments on the relative importance of the three management strategies).

Thank you very much for your suggestion and we fully agree with you. In the revised manuscript we no longer highlight the relative benefits of different measures, but instead call for a combination of different measures which are more effective than single measure implementation. In the discussion we added, to acknowledge the challenge of parameter selection:

“there are challenges in accurately parameterizing the models, thereby affecting to some extent the relative importance of measure, which can also be dependent on model assumptions” (see lines 529-531)

Responses to Reviewer 2

Thank you for your constructive comments for our paper. After making modifications based on your comments, we have greatly improved the details of the manuscript. Below, we indicate how we responded to each of your comments.

[R2C1] The paper has important conservation value in policy level decision making and it is written relatively well with appropriate analysis.

Thank you for your kind words about the manuscript and your encouragement for our work.

Abstract

[R2C2] Line 35, Change “small-isolated threatened population” to small-isolated threatened population

Agreed. We have modified it according to your comment.

Introduction

[R2C3] There is no rationale explained that how habitat expansion can influence impact of CDV. Also, the manuscript highlights the dogs as potential carrier but fails to explain the action/ mode of transmission of infection from dog to wild animals. Overall reason for initiation of this study can be strengthened.

This paper focuses on modeling the effects of CDV control (in two ways: dog control and leopard vaccination) and habitat expansion on leopard populations. Habitat expansion is a separate measure that aims to increase the carrying capacity for the population and is not intended to slow the effects of CDV, however, for sake of scenario comparison, we prefer to include it, as it gives to the reader important information about the limiting factors affecting the leopard population.

To make it clearer, we have added the following

“Alternative (A) and (B) represent measures directed towards the control of CDV, while for alternative (C), although habitat expansion can also indirectly reduce the transmission of CDV to some extent, in this study, habitat expansion is aimed at increasing the carrying capacity for the population, not at mitigating the effects of CDV.” (line 290-293)

The transmission of CDV from dogs to leopards is mainly through leopard predation on dogs, but of course, in addition to dogs, leopards can also be infected with CDV through predation on other small carnivores, as we explain:

“The leopard can prey on various virus hosts (free-ranging domestic dogs and small sized carnivores such as the Asian badger *Meles meles*, red fox *Vulpes vulpes*, and leopard cat *Prionailurus bengalensis*) that act as infection pools.” (see lines 69-71)

This is why dog control alone cannot completely eliminate the spread of CDV in leopard populations, which is why we have carried out a simulated CDV vaccination of leopards.

In addition, we explain the two ways of CDV transmission to leopards:

“Transitions from susceptible to exposed occurred in two ways: one is from the environment, *i.e.*, by predation on domestic dogs and small wild carnivores; the other is the interactions with other infectious leopards” (lines 166-168)

[R2C4] Line 97, Don't know how appropriate to use land conditions which this context. And what are the land conditions can be explained further. May be reworded

Sorry for the ambiguity. We refer to the habitat in terms of area, therefore we changed the “land conditions” to “area conditions”

[R2C5] line 100 Remove “to” from there is no research to available

Agreed. We have modified it, thank you for finding this mistake.

[R2C6] Line 105 -108, Here it has been mentioned that Vortex model is very comprehensive but authors have shown only inbreeding depression as component of the model. To assert its comprehensiveness all important component of the model should be highlighted.

Vortex incorporates density constraints, genetic factors, spatial dynamics and other functions in addition to inbreeding depression, but not all of these functions were applied in this study, for example, density constraints and spatial dynamics were not considered in this study. The sentence to which this comment is referred is intended to emphasize the importance of the inbreeding depression function contained in Vortex for this study. So, to avoid confusion, based on your suggestion, we have revised this paragraph to:

“Vortex software package has a wide range of applications in PVA, which has been used to evaluate the survival status and different management strategies for the conservation of endangered species, such as the griffon vulture (*Gyps fulvus*), fennec fox (*V. zerda*), and mountain lion (*Puma concolor*) (Aresu et al. 2021; Benson et al. 2019; Franklin et al. 2021). In addition, Vortex incorporates inbreeding depression (the organism's reduced ability to

survive as a result of the inbreeding of related individuals), which is very important for small-sized populations like the object of our study.” (lines 94-100)

[R2C7] Line 128-129, Not clear. Rewrite the sentence for clarity.

Thank you for your comments. We rewrote this way:

“The results of our analysis can inform the management policy for the Amur leopard's persistence.” (see lines 116-117)

[R2C8] In Materials and Methods, line 136, or cubs individuals – how many?

The nine individuals included subadults and cubs. To express more clearly, we modified it to “78 adults and 9 subadults/cubs” (line 122) (refer to R1C10)

[R2C9] Line 140, Change free-range domestic dogs to free-ranging domestic dogs

Thank you for your comment, we changed all “free-range” to “free-ranging” in the new version of the manuscript.

[R2C10] The abbreviation (LEs) given to Lethal equivalents and Inbreeding depression. Please recheck and correct it.

We gave the abbreviation (LEs) to Lethal Equivalents, and we modified the Inbreeding depression in Table 2 to Lethal equivalents. Thanks.

[R2C11] Line 213-216, Before mentioning this, effect of population size and encounter rate can be highlighted with citations.

Since Amur leopards are solitary animals. After our discussion, we feel that the effect of changes in population size on the transmission of CDV in Amur leopard population is limited, and this effect is not considered in this study. And the increase in encounter rate certainly accelerates the transmission of CDV in the leopard population, which is common sense, so we did not put extra emphasis on it, but made a reasonable estimation of the encounter rate based on the life habits of Amur leopards.

Discussion

[R2C12] How realistic is your viability analysis metamodel? Need more details about its application for revival of threatened wildlife population in discussion which is more practical application oriented.

Of course, the metamodel is not predictive of the real world, as pointed out by Reviewer 1 in R1C2 and R1C15. The results are oriented and serve to give a

prediction in terms of relative importance of different management strategies. The important is that limits are acknowledged and results are discussed respecting the assumptions made.

Also like Reviewer 3 said in R3C1: “This study can be a tool for making decisions with imperfect knowledge of the system state of wildlife population, especially on limited timelines.” And we can explore more deeply in the next work like the Reviewer 3 suggested in R3C4

To make this point more clearly, we have adjusted section 4.3 of the discussion as follows:

“We have demonstrated the theoretical and practical application of a metamodel combining the uncertainties of epidemiology and demography in an individual-based context. Although the future trends of the population obtained from our simulation analysis can provide good guidance for management measures to maintain the sustainability of the Amur leopard population, there are challenges in accurately parameterizing the models, thereby affecting to some extent the relative importance of measure, which can also be dependent on model assumptions. Our results should be viewed as demonstrating the relative benefits of various possible measures rather than as absolute, accurate predictions of future population trends.” (lines 525-533)

[R2C13] Line 463-466, It is better to give probable reason for inconsistency in the model.

Since the results have changed a bit due to rerunning the model (see R1C3 and R1C4 for details), the inconsistency is now absent and, consequently, we removed the description.

[R2C14] Line 493-494, Other potential strategies to mitigate inbreeding can be considered mentioning in this paragraph

Thanks for the suggestion, we agree.

We added, at the end of the paragraph.

“In addition, habitat expansion can also reduce inbreeding depression to some extent.” (see lines 491-492)

[R2C15] Line 508-509, Measures to ensure protection can also be highlighted

Agreed, we added, at the end of the paragraph:

“And it is vital to establish strict management policies to prevent poaching of ungulates.”
(see lines 514-515)

[R2C16] The manuscript should explain/suggest the way forward and follow up actions required based on your core findings. How your findings are going to help in minimizing further decline of species.

Thanks for this suggestion:

We wrote:

“In summary, the conservation of the small Amur leopard population requires a multifaceted effort, including improving the survival environment of females to enhance their reproduction and survival rates, as well as considering the disturbance caused by CDV epidemics. Finally, increasing the management and vaccination of free-roaming domestic dogs in the forest, vaccinating leopards and gradually expanding the range of the habitat, are all strategies that should be thoroughly considered timely by wildlife. When the inbreeding depression is severe, the synergistic consideration of the various management actions is critical.” (see lines 516-523)

[R2C17] In Table 1 and Table 2, you need to give references for all your baseline values. You can create another column and add the references.

Thank you very much for your comments. We understand that this would be practical. However, most of the parameters used are not directly cited from other articles, but are obtained by secondary calculations using data provided by other articles. For this reason, it is quite complicated to fill the table in a synthetic way. Respectfully, we prefer to use the text and the Supplementary Information to explain the detailed process of calculations, and the references used. In addition, some parameters were reasonably estimated and did not have references, such as the probability that an individual never becomes susceptible, the transmission probability from an infectious mother to a newborn, the time that maternally derived immunity protects an offspring. The value of these parameters are based on reasonable estimation from the knowledge of the species life-history traits.

Responses to Reviewer 3

Thank you for your comments concerning our manuscript. Your comments are all valuable and very helpful for revising and improving our paper, as well as the important guiding significance to our research. Below, we indicate how we responded to each of your comments.

[R3C1] Population viability analysis (PVA) is a general tool for assessing viability of the endangered population. The author used PVA to assess the effectiveness of different management conservation measures on the critically endangered Amur leopard population under various inbreeding depression scenarios. This study can be a tool for making decisions with imperfect knowledge of the system state of wildlife population, especially on limited timelines. Although the PVA requires numerous parameters and long term monitoring work, which increases the uncertainty of the model to some extent, this study gave detailed parameter estimates and sources of the required parameters in the Supplementary Information, which made the model reproducible. In terms of method, the author used a novel PVA metamodel incorporating CDV epidemiology to provide practice guidelines for Amur leopard conservation, which has important insights for small populations of endangered species.

Thank you very much for your recognition and encouragement for our research

[R3C2] In the sensitivity analysis, for the 6 parameters, such as Mortality after CDV infection (MCI), values plus or minus 40% of the baseline values were tested. Why 40% was chosen instead of other values, please explain.

Thanks for this comment, we agree that this might be somehow arbitrary. We chose $\pm 40\%$ for two main reasons: (1) the $\pm 40\%$ range of variation has been used in published articles (He et al. 2020), and we have added this reference in the text. (2) we think this range of variation covers the possible parameter changes in real situations.

In the text we have added the following:

“A total of 16 simulations were carried out: with test values for LEs of 0 and 6.29, CCI of 3 and 7 years and for the other 6 parameters, values plus or minus 40% of the baseline values were tested⁵⁸, and this range of variation covers the possible parameter changes in real situations.” (lines 320-322)

There are indeed challenges regarding the selection of parameters, and we have added a relevant discussion:

“Although the future trends of the population obtained from our simulation analysis can provide good guidance for management measures to maintain the sustainability of the Amur leopard population, there are challenges in accurately parameterizing the models, thereby affecting to some extent the relative importance of measure, which can also be dependent on model assumptions” (line 585-589)

Please, also consider that the metric used as sensitivity index is normalized by the parameter change. This should limit the arbitrariness of the choice of the parameter.

[R3C3] In the methods section, especially the model description (CDV epidemiological model description, Demographic model description), takes up too much space and could be considered to be shortened or moved partly to the Supplementary Information.

Thank you very much for your comments. Respectfully, we prefer to keep the description of the methods as is. Before submitting the first version of the manuscript, we had already made choices and some detail is in the Supplementary Information. And we believe that the information provided in the text is necessary for a complete high-level description of the models. However, we have tried to shorten this section as much as possible. Starting from 1668 words, arriving to 1250 words, reduced by 418 words.

[R3C4] The dog control measure is optimal when the three conservation measures are implemented individually, and this should be clearly stated in the discussion and possibly do some work to test it in next work.

Since we modified the parameters on the probability of CDV infection in dogs and corrected errors in the model, in the new results, dog control obtains approximately the same result as the benefit of vaccinating 4 leopards per year, while the benefit of vaccinating 6 leopards per year is better than dog control, so the comparison of relative benefits between measures here depends on the number of leopards vaccinated per year. So, we do not overemphasize the relative benefits of the measures (please refer to [R1C2]). Also thank you very much for providing us with research ideas for the next step of exploring in more detail the impact of domestic dogs on wildlife.

[R3C5] Note the use of italics in the main document, e.g. in line 327.

Thanks, we have corrected it and we have been consistent throughout the new version of the manuscript.

[R3C6] The paragraph in line 433, and the paragraph in line 463 are both CDV control measures, but the paragraph in line 449 is a habitat expansion

measure, so it is suggested to exchange the position of the paragraphs in lines 463 and 449 to ensure the coherence of the content.

Thank you for your valuable comments. We have adjusted the paragraphs to make the content of the article coherent.